# The quantitative genetics of gene expression in Mimulus guttatus

**Paris Veltsos**[¤], **John K. Kelly**[*]

Department of Ecology and Evolutionary Biology, University of Kansas, Lawrence, Kansas, United States of America

¤ Current address: Ecology, Evolution and Genetics Research Group, Biology Department, Vrije Universiteit Brussel, Brussels, Belgium
* jkk@ku.edu

## Abstract

Gene expression can be influenced by genetic variants that are closely linked to the expressed gene (cis eQTLs) and variants in other parts of the genome (trans eQTLs). We created a multiparental mapping population by sampling genotypes from a single natural population of *Mimulus guttatus* and scored gene expression in the leaves of 1,588 plants. We find that nearly every measured gene exhibits cis regulatory variation (91% have FDR < 0.05). cis eQTLs are usually allelic series with three or more functionally distinct alleles. The cis locus explains about two thirds of the standing genetic variance (on average) but varies among genes and tends to be greatest when there is high indel variation in the upstream regulatory region and high nucleotide diversity in the coding sequence. Despite mapping over 10,000 trans eQTL / affected gene pairs, most of the genetic variance generated by trans acting loci remains unexplained. This implies a large reservoir of trans acting genes with subtle or diffuse effects. Mapped trans eQTLs show lower allelic diversity but much higher genetic dominance than cis eQTLs. Several analyses also indicate that trans eQTLs make a substantial contribution to the genetic correlations in expression among different genes. They may thus be essential determinants of "gene expression modules," which has important implications for the evolution of gene expression and how it is studied by geneticists.

## Author summary

Mimulus guttatus (yellow monkeyflower) is a model for the study of quantitative trait evolution in natural populations. Research has focused mainly on whole organism traits like flower size or herbivore resistance, but the level of expression of a gene is also a quantitative trait. In this study, we dissect leaf transcriptome variation using a breeding design that estimates the contribution of individual loci to expression variation (eQTLs). We find rough agreement to the "oligogenic model" of inheritance where a major locus (the cis regulatory region) generates much of the genetic variation in the population. Associations studies usually characterize genetic effects as binary (e.g. the two alternatives at a single nucleotide polymorphism or "SNP"), but this description is insufficient for Mimulus.

**Data Availability Statement:** The trimmed RNAseq reads have been submitted to the Sequence Read Archive (SUB12286589, SUB12291949) under bioproject PRJNA902708.

**Funding:** This research was funded by NSF grant MCB-1940785 to John Kelly. Paris Veltsos and John Kelly received salary support from this award. The funders had no role in study design, data collection and analysis, decision to publish, or preparation of the manuscript.

**Competing interests:** The authors have declared that no competing interests exist.

Most loci exhibit multiple, and in some cases, a continuum of alleles. We find that trans eQTLs have different features than cis eQTLs, both in terms of the diversity and genetic dominance of alleles. These genetic features of eQTLs are critical determinants of the "G matrix," the genetic variances and covariances among all genes which determine how gene expression will evolve under selection in response to changing environmental conditions. Our finding of large effect sizes and high allelic diversity suggests that the G matrix may be surprisingly malleable, even on ecological timescales.

## Introduction

Gene expression is a quantitative trait. Expression scored from sequence-read counts (RNAseq [1]) is strongly influenced by environmental variables, measurement error, and the complex interaction of many genes [2]. A vast methodology has been developed for the analysis of quantitative traits with applications to agriculture, conservation, and the evolution of natural populations [3,4]. When RNAseq is applied to a population, specifically to a collection of genotypes that have been randomly sampled from a deme, the machinery of quantitative genetics can be employed to address basic questions about the potential for gene expression to evolve. We can ask how many loci affect expression of each gene and how large their respective effects are. The effect/number distribution is essential for predicting how rapidly expression will evolve under natural selection [5,6]. Population allele frequencies are a second critical aspect of quantitative trait variation. Determining whether alleles at expression Quantitative Trait Loci (eQTLs) are typically rare or intermediate in frequency tests hypotheses about the evolutionary forces that maintain variation [7]. Next, we can ask whether gene expression is affected by genetic complexities such as dominance, epistasis, or genotype by environment interaction. These factors influence the mapping from genotype to fitness and thus the amount of genetic variation in expression available to selection. Finally, recognizing that the entire transcriptome is just a very long vector of quantitative traits [8], we need to determine the genetic basis of correlations among genes in their expression levels. Estimating the respective contributions of genetic and environmental factors to covariances is essential to understanding co-expression patterns across the genome.

Gene expression is unlike other quantitative traits in that we know the location of one very important locus prior to genetic mapping. The DNA surrounding a gene is likely to contain regulatory sequences such as promoters and enhancers. This locus, the cis eQTL, is thus a strong candidate as an effector of expression. What fraction of the total genetic variance in expression is generated by the cis eQTL? The proximity of regulatory DNA to the expressed gene suggests an oligogenic model of inheritance [9], where most variation is generated by a "major effect" cis eQTL. There will also be a lesser contribution of numerous unlinked modifiers (trans eQTLs). However, association studies of gene expression variation in humans suggest a very different model. Even if the cis eQTL is the most important single locus, it may explain only a minor fraction of the genetic variance in expression. The omnigenic model [10,11] posits that many trans eQTLs, each with small effects and distributed uniformly over the entire genome, generate the bulk of variation in expression.

Genetic dominance is likely to differ between cis and trans eQTLs. Additive gene action is expected for cis eQTLs [12,13] given that regulatory molecules like transcription factors bind separately to each allele. With allele-specific effects on expression, additivity results if the overall expression of a gene is the sum of the mRNAs produced independently by each allele. This simple model can breakdown if there is imprinting [14] or if feedback mechanisms such as

autoregulation [15] cause the realized mRNA levels of one allele to depend on the expression of the other. In contrast to cis, there is no *a priori* reason to assume additive gene action for trans eQTLs. The product of a trans acting locus (say a transcription factor protein) can affect both alleles of the expressed gene [2].

Cis and trans eQTLs should also contribute differentially to genetic covariances between expressed genes. Genetic covariances result from pleiotropy, linkage disequilibria, and in populations that inbreed to some extent, identity disequilibrium [16,17]. In this paper, we apply a breeding design where all individuals have a known ancestry. This allows us to estimate the combined effects of pleiotropy and LD on the co-expression of genes and the contribution of individual QTLs to these covariances [18,19]. When considering multiple expressed genes, a single locus can have multiple effects, both cis and trans. While it is typical to think of cis eQTLs as effectors of a single gene, a single mutation could affect the expression of multiple closely linked genes by altering local DNA accessibility. Distinct mutations in regulatory regions of closely linked genes will generate a genetic covariance if these mutations are in linkage disequilibrium in the population. Trans eQTLs can generate genetic covariances in several ways. Most obviously, a trans eQTL that affects many genes will generate covariation in expression among all its targeted genes. More directly, the cis effect of a mutation on a regulatory gene should generate a correlation between the expression of that gene and the expression level of downstream target genes (for which the mutation would be a trans eQTL).

In this paper, we describe an experiment to characterize variation and covariation in gene expression, and then estimate the contribution of individual genetic loci to this (co)variation. We created a multiparental mapping population by intercrossing genotypes from one natural population and then measured gene expression in leaf tissue (Fig 1B). The replicated $F_2$ crossing design (Fig 1A) produces high variance in relatedness of individuals, which is essential for estimating genetic (co)variances. It generates both homozygous and heterozygous genotypes at individual loci, necessary for characterizing how both the additive and dominance effects of eQTLs contribute to variation. We analyzed these data using two complimentary approaches. The "Cross-specific analysis" treats each of the nine families as a distinct entity and extracts estimates for QTL effects in the fashion of a single $F_2$ mapping population, e.g. [20]. The "Combined analysis" considers all plants simultaneously with the relatedness of each $F_2$ plant to all other plants estimated through genomewide similarity [21]. Given sufficient variation in relatedness, we partition expression variation into genetic and environmental components using the classical "animal model" ([22], i.e. the linear mixed model [23]). Finally, we determine the contribution of individual loci to the genetic component of variation established in this context.

## Results

### Mapping RNAseq reads to our *de novo* assemblies effectively genotypes F2 plants

Two of the ten parental lines (767 and 62) used in this study were sequenced and assembled by the Joint Genome Institute [24], while the other eight were assembled from our long-read sequencing (see Methods A). *De novo* assembly of the long reads yielded two to four large scaffolds per chromosome with a high inclusion of genes (BUSCO completeness 93–94%, S1 Table). We used genetic maps obtained from our $F_2$ genotyping to assemble scaffolds into pseudo-chromosomes. Next, we called SNPs among these lines and report the nucleotide diversity within and around each gene in S2 Table. These comparisons confirm our previous Illumina sequencing [25]: The 10 lines are about equally distant from each other in terms of

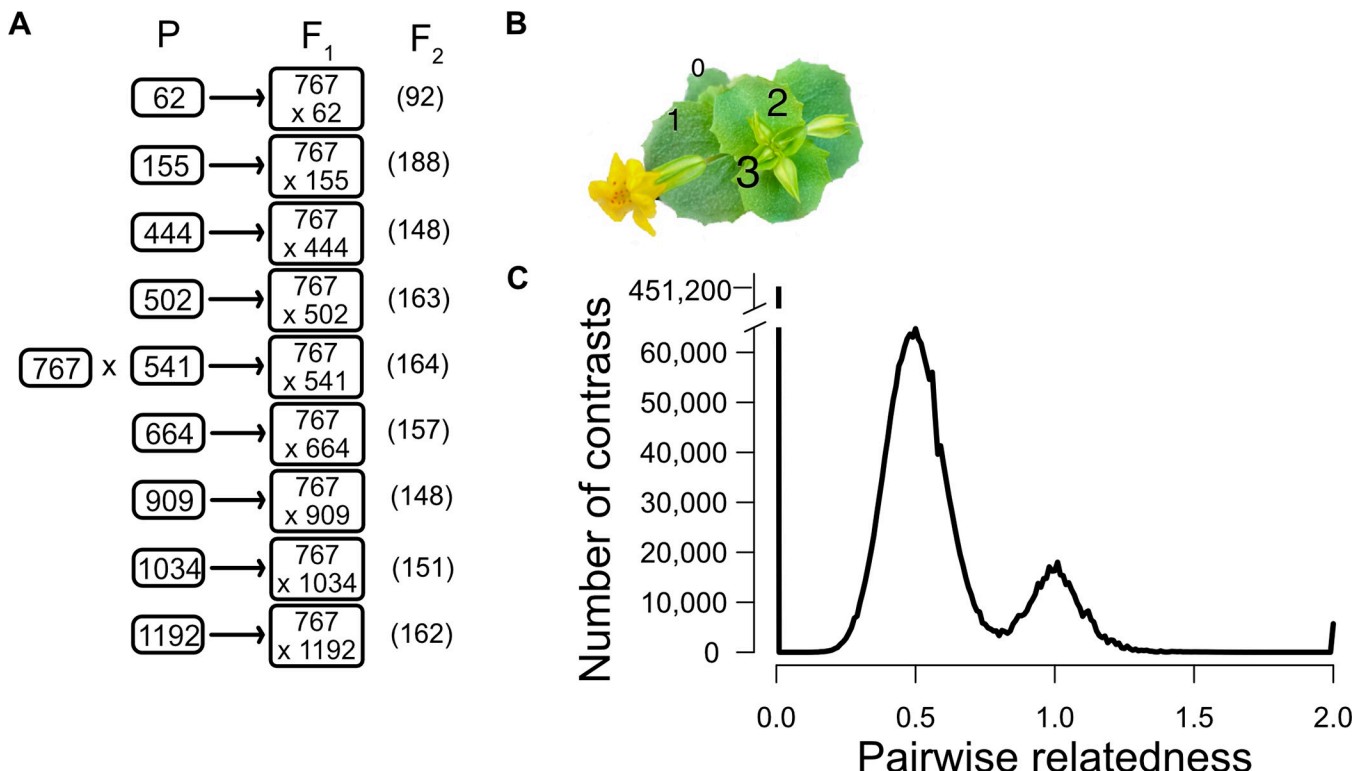

**Fig 1.** (A) A diagram of the replicated F2 design with the number of plants used after filtering in parentheses. Each "P" is an unrelated inbred line. 767 is another inbred line derived from the same population, unrelated to the other nine P lines. (B) A photo of the plant (leaf number noted) at the developmental stage when $2^{nd}$ leaves were harvested. (C) The distribution of relatedness (twice the coefficient of coancestry) values from all pairwise comparisons of individuals. The set of contrasts centered on 0.5 corresponds to $F_2$ individuals of different families, while the contrasts centered on R = 1.0 come from intra-family comparisons. Comparisons among plants of the same inbred line have R = 2.0 (genetically identical and fully homozygous).

sequence divergence (S1 Fig) and can thus be treated as unrelated individuals from the natural population.

We genotyped $F_2$ plants using the RNAseq reads (Methods C). Transcript reads can be sub-optimal for genotyping owing to varying coverage per locus (expression levels differ among genes) and because the representation of the two parental alleles within the RNA of heterozygotes may be unequal (allele-specific expression). We address these difficulties by stringent filtering of genes, using only about 37% as genetic markers. Next, we apply a Hidden Markov Model (HMM) to each chromosome of each individual allowing marker specific genotyping error rates (the emission probabilities of the HMM). The HMM leverages genetic information across the chromosome, and particularly from neighboring genes, to call the genotype (ancestry) at each locus [26,27]. Given the recombination rate of *M. guttatus* [27], a diploid $F_2$ plant has an average of ~1.8 crossover events per chromosome. Consequently, there are large stretches of markers (usually hundreds of genes) between genotype transitions along chromosomes. Neighboring markers will (nearly) always have the same ancestry (homozygous for 767, heterozygote, or homozygous for the alternative parental line allele), which greatly simplifies genotype inference. For the filtered dataset, we obtained posterior genotype probabilities of >99% at virtually all marker loci.

After filtering the RNAseq read data, we obtained an average of 4,800 informative genetic markers per cross (family). The HMM yields genetic maps for each family. The maps from different families are similar to each other by chromosome, and the average total genetic length (1,260 centimorgans) is comparable to previously published maps of *M. guttatus* [27]. Also,

the maps exhibit the predicted pattern of recombination suppression over regions where large inversions are known to segregate. Line 664 carries a large inversion on chromosome 6 [28] and the map for this family exhibits recombination suppression over the predicted region (1.22 to 8.57mb in the coordinates of the 767 genome assembly). The meiotic drive allele on chromosome 11 [29] segregates in families 62, 502, 541, 664, and 909, and these maps exhibit consistent suppression from coordinates 6.60mb to 17.62mb. As expected, this interval includes the centromere. Interpolating from the genetic markers, we established a nearly complete genotype matrix for eQTL mapping. For each gene measured for expression, we could score the locus as homozygous for the reference line (767), homozygous for the alternative line, or heterozygous. This gives us a genotype call specific to each expressed gene, which is the cis eQTL in the Combined analysis.

## Estimation of genetic variances and the contribution of specific eQTLs

Genetic variances are estimated by determining how phenotypic similarity increases with genetic similarity. Estimation will be most effective when we can compare plants that range from unrelated (R = 0) up to fully homozygous identical twins (R = 2). We calculated the relatedness using the genotype matrix for the 1,588 plants. The distribution of pairwise relatedness values (depicted in Fig 1C) confirms that our crossing design produced the high variance in relatedness that is necessary for accurate estimation of the genetic variances. At the low end, where R = 0, there are 451,200 contrasts among unrelated individuals. These are plant pairs from different inbred lines and contrasts of $F_2$s to unrelated parental lines. The next most frequent contrast is among $F_2$ plants in different families (average R = 0.5), which are related through shared inheritance of DNA from their common parent (767). $F_2$ plants within a family (average R = 1.0) can have alleles Identical by Descent through both parents. The variability in relatedness around the modal points of 0.5 and 1.0 are due to the stochastic events of segregation and recombination in gamete formation that will make siblings more/less similar by chance. Finally, there are several thousand contrasts of genetically identical individuals within parental lines. These contrasts have R = 2.0 because line plants are completely homozygous (the maximum for R is 1 when all plants are outbred).

We used simulations to choose the best statistic to estimate the contribution of individual loci to genetic variation in expression (Methods F). In these simulations, we used the observed genotype matrix as a framework with subsequent generation of expression levels, with and without cis and trans eQTLs of varying effects. We tested the accuracy of three different methods: the least squares based Haseman–Elston (HE) regression [30] and two statistics derived from the maximum likelihood fit of the linear mixed model: Vg[r2] and Vg[a] are described in Methods D. Across a range of cases, all three estimators are nearly unbiased given our sample sizes and genotype matrix. In other words, the average of estimates across simulations is close to the true value of the parameter used to simulate data. However, when a locus contributes to genetic variation, the variance among replicate simulations is much larger for HE regression than for either Vg[r2] or Vg[a]. The variance for Vg[a] is marginally lower than for Vg[r2] (see S3 Table). We chose Vg[a] to estimate the contribution of both cis and trans eQTLs to the genetic variance of expression in the real data because it was the most accurate (Vg[a] has the lowest mean square error).

## The great majority of genes exhibit cis-regulatory variation with high allelic diversity

In the Combined analysis, 91% of genes have a significant cis eQTL (FDR < 0.05; S4 Table). The Cross-specific mapping identified 32,853 eQTLs over the 9 families, most (22,794) were

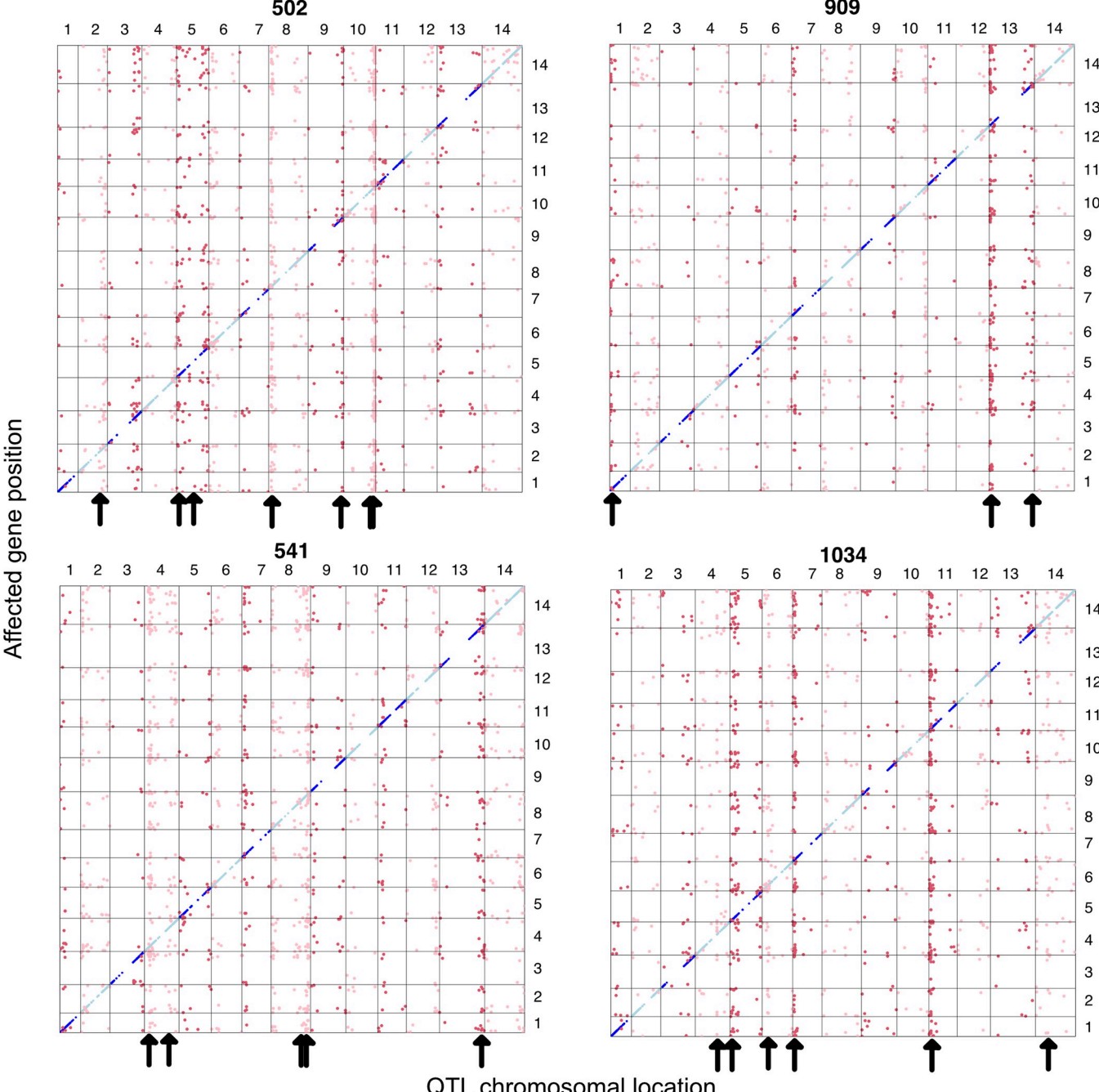

**Fig 2. All significant eQTLs are reported by QTL (x-axis chromosomal locations) and affected gene position (y-axis) in four of the crosses (alternative lines 502, 909, 541, and 1034).** Blue/aqua points are cis eQTLs while red/pink denotes trans eQTLs (shade changes with chromosome). The vertical "chimneys" highlighted by arrows are trans eQTL hotspots, the locations of which are unique to each cross.

cis to the affected gene. These estimates refer only to measured genes, those expressed in leaf tissue and passing filters. The eQTL plot for four of the crosses (Fig 2) are typical of the full set (S2 Fig), with the many cis eQTLs filling the diagonal of this plot. Gaps along the diagonals are only present in centromeric parts of chromosomes where there are few genes. This is partly due to filtering: We did not test genes with a mean Count Per Million < 0.5. Average

expression levels are lower for genes in centromeric regions (and for low recombination portions of the genome generally; S3 Fig). A greater number of significant cis eQTLs emerge from the Cross-specific analyses than the Combined analysis (22,794 > 11,818) but these numbers are not comparable. Each gene is tested nine times in the former analysis (once within each cross) but only once in the latter.

Genetic effect estimates are very similar between Cross-specific and Combined analyses. When cis QTL effects are measured in standard deviations of the expression level, estimates for the same locus/cross are nearly equivalent between identical Cross-specific and Combined analyses: The correlation is 0.96 when including both significant and non-significant tests (n = 9 crosses x 12,987 genes = 116,883 estimates; S4 Fig). This high congruence is noteworthy given that (a) data transformations differ between pipelines, (b) the Cross-specific analysis considers only $F_2$ individuals while the Combined analysis also includes data from the homozygous parental lines, and (c) the Combined analysis includes a random effect to absorb trans eQTL effects while the Cross-specific does not. The strength of evidence (level of statistical significance) for cis-regulatory variation is much stronger from the Combined analysis because it integrates signal across families. However, the point estimates for allelic effects are remarkably consistent (r = 0.96).

More important than the number of significant tests, we find that most of the additive genetic variation in gene expression is explained by cis eQTLs. From the Combined analysis, the mean values for $V_E$, $V_{g(cis)}$ and $V_{g(trans)}$ were 0.828, 0.093, and 0.044, respectively. The fraction of the genetic variation generated by the cis locus varies (Fig 3A), but $V_{g(cis)} > V_{g(trans)}$ for 63% of genes. Here, $V_{g(trans)}$ is estimated using the relationship matrix and represents the combined effect of all trans acting loci on the affected gene. As expected, the strength of evidence for a cis eQTL is positively correlated with $V_{g(cis)}$ (S5 Fig). The residual variance, $V_E$, is the largest component for most genes where measurement error owing to finite sequencing depth contributes substantially. Average read depth per gene is a strong positive predictor of test significance, while average $V_E$ declines as coverage increases (S6 Fig). Both trends are expected if expression levels are less accurately estimated at genes with lower coverages.

The cis eQTLs identified by our Combined and Cross-specific analyses are significant when the overall expression of a gene differs among the three genotypes that segregate within each family (the alternative homozygotes and the heterozygote). Because genotype is called at the marker locus most proximal to the expressed gene, significant tests from this procedure are often called "local eQTLs" [31]. Allele-specific expression provides an alternative method to detect cis eQTLs [32] based only on data from heterozygous individuals. If cis DNA variation only affects the expression of the physically linked gene (on the same chromosome), the "high allele" should be over-represented in the mRNA produced by heterozygotes. In this experiment, we have the genome sequences of all parental genomes and can distinguish alternative alleles within the mRNA produced by heterozygous plants for 46,828 gene/family combinations. In these cases, we see that whichever allele increases expression across all three genotypes is usually over-represented in the mRNA produced by heterozygous plants (Fig 3B). This is the signature of allele-specific expression. Given that both estimates (the additive effect on overall expression and allele frequency within heterozygotes) are subject to substantial estimation error, the high positive correlation (r = 0.83) provides a compelling corroboration of cis eQTLs identified using local markers. This is illustrated by the subset of cases where the estimated additive effect of the alterative allele on expression is lower than -0.5 or greater than 0.5 (and we can be confident of correctly inferring up or down regulation). In these cases, allele frequency in the mRNA of heterozygotes deviates from 0.5 in the predicted direction in 99.6% of 7,841 tests.

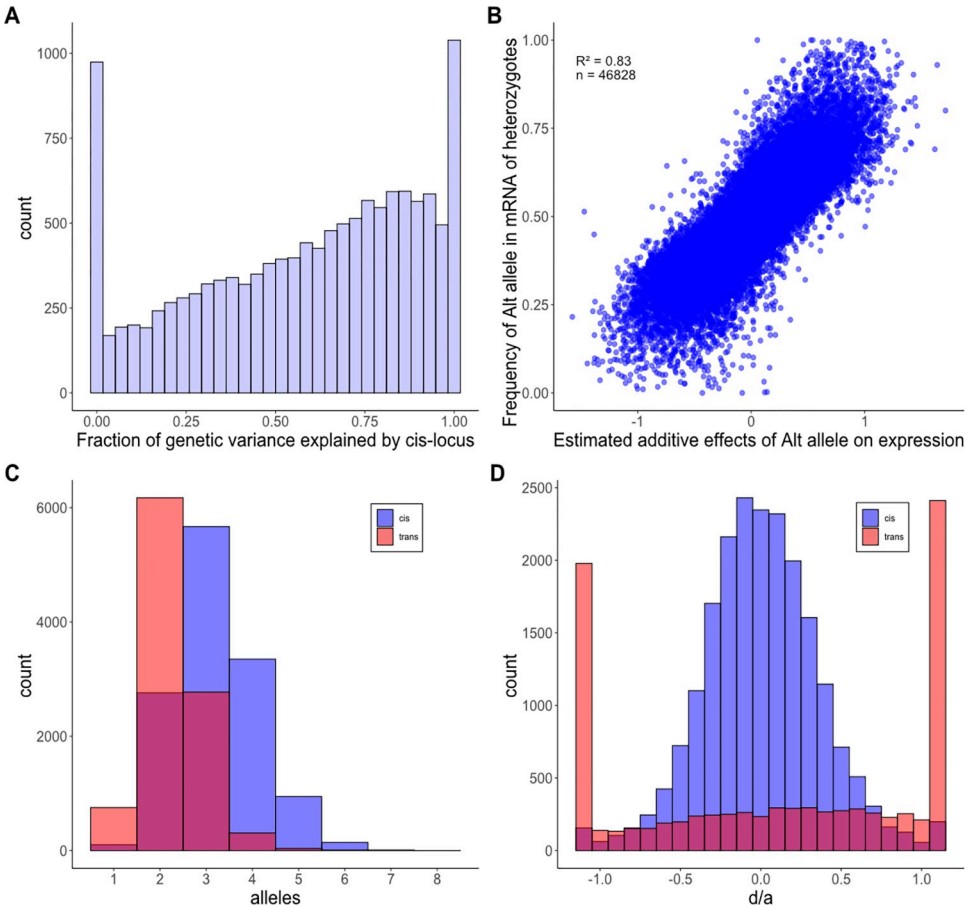

**Fig 3.** (A) The distribution (across genes) of the fraction of the total genetic variance due to the cis locus is reported for each of the 12,987 genes. Genes with negative Vg(cis) estimates are reported as 0. (B) The estimated effect of the alternative allele (not 767) on total gene expression (across genotypes) is a strong predictor of allele frequency within the reads produced by heterozygotes. The units for a (estimated additive effect) are standard deviations of expression. (C) The distribution across eQTLs of the number of functionally distinct alleles per cis (blue) and trans (red) eQTL are reported. (D) The distributions across genes of estimates for the scaled dominance coefficient (d/a) are reported for cis and trans eQTLs. With no dominance d/a = 0 while -1 (or 1) implies that the alternative allele is fully recessive (or dominant). The end categories bin all estimates that are less than -1.1 or greater than 1.1.

Contradicting the assumption of additivity, about 20% of cis eQTLs exhibit some level of dominance (the test for dominance yields FDR<0.05). However, dominance is nearly always partial with heterozygote expression levels between the values of alternative homozygotes. Dominance is quantified by the parameter $d$, where the mean expression of genotypes RR, RA, and AA are $m$, $m+a+d$, and $m+2a$, respectively ($m$ is the mean expression of individuals homozygous for the reference allele (RR)). Partial dominance is implied if abs($d$) < abs($a$). For cis eQTLs, 98% of point estimates for $d$ and $a$ satisfy this condition (Fig 3D). Partial dominance at cis eQTLs is corroborated by the allele-specific expression data. If we regress the alt-allele frequency in reads from heterozygotes (y-axis of Fig 3B) onto the estimates for both $a$ and $d$ simultaneously, both are highly significant as positive predictors of allele frequency ($p < 10^{-9}$ for each coefficient). The positive coefficient for $d$ means that when the overall expression of the heterozygotes exceeds the value predicted by additivity, there is a corresponding increase in alt-allele frequency within the reads produced by heterozygous plants. When $d < 0$, this

frequency is reduced relative to additivity. Thus, fluctuations in allele-specific expression parallel dominance estimates.

The definition of "alleles" is different in multiparental mapping experiments (such as this study) than in genomewide association studies. In the latter, alleles are typically biallelic SNPs. Here, alleles are the distinct haplotypes carried by the founding parental lines in the vicinity of each gene. In our initial model fit, we allowed the cis allele from each of our nine alternative lines to uniquely differ from the allele carried by the reference line (767). From the point of view of statistical testing, it is appropriate to allow each allele to have a unique effect on expression that is characterized by a distinct free parameter. In fact, our simulations indicate this procedure to be slightly conservative for detecting QTLs (Methods F). However, in terms of characterizing QTL effects, this "full model" is overparameterized when fewer than 10 functionally distinct alleles segregate. To address the number of functionally distinct cis alleles, we applied the allele partitioning method of King *et al.* [33] to each gene. The typical result is an allelic series with a median of 3 alleles per cis locus (Fig 3C). Some significant loci have only two distinct alleles, but in this case, each allele is typically carried by multiple ancestral lines. We can characterize allele number and relative frequency of alleles with heterozygosity: $H = 1 - \sum q_i^2$, where $q_i$ is the frequency of the $i^{th}$ allele and the sum is taken over all alleles at the locus. Across all cis loci (significant or not), the median H is 0.59 indicating high allelic heterogeneity.

We used multiple linear regression to test if sequence variation in the vicinity of genes can predict the strength of cis eQTL effects on expression. We measured variation within each of three windows around each gene: the 1kb upstream of the gene start codon, the gene itself, and the 1kb downstream. Within each region, we calculated nucleotide diversity ($\pi$) and a measure of insertion/deletion frequency (U) as potential effectors of cis eQTL significance. U = 0.0 if the ten lines are perfectly colinear over a region but increases toward 1.0 as indels accumulate (see Methods B). Predicting the significance of cis eQTL effects on the full set of 12,897 genes, all six predictors are positive, but only four are strongly significant (S5 Table). Cis eQTL significance increases most strongly with $\pi$ within the genic region and with U for the upstream regulatory region. Indels in the genic and downstream regions have moderately positive effects on cis eQTL significance, while nucleotide diversity in the upstream and downstream regions are minimally important.

## The characteristics of ascertained trans eQTLs are very different from cis eQTLs

While less frequent than cis eQTLs, trans eQTLs are abundant: 10,059 significant tests across all nine families in the Cross-specific analysis (Fig 1). A substantial fraction of trans eQTLs occur in "hotspots" where a single locus effects the expression of many genes. Within each cross, we clustered significant trans eQTLs if located within 2 centimorgans of each other which distills all significant tests into 1,979 loci (S6 Table). The number of affected genes per locus is usually low (median = 2), but we can identify 35 hotspots where the trans eQTL affects the expression of 30 or more genes. In a few cases, hotspots in different families have roughly similar genomic locations (S2 Fig), but since the affected genes are different, they are likely caused by different mutations. For example, three different families (155, 664, and 909) each have a trans eQTL hotspot within the first Mb of chromosome 1. However, the 68 affected genes in family 155 are different from the 30 genes affected in family 664, and the 35 in family 909 are non-overlapping with either previous set.

In the Cross-specific analysis, 98% of the trans eQTL/affected gene pairs were ascertained within only one family. This suggests low allelic diversity–a bi-allelic polymorphism with the

minor allele carried by only one parental line. However, it is not tenable to assume the absence of significance as absence of effect. To provide a meaningful contrast of allelic diversity at cis and trans eQTLs, we estimated the effect of each trans eQTL considering all families simultaneously, essentially applying the Combined analysis model previously fit to cis loci. Across all trans eQTL/affected-gene pairs, the Combined analysis indicates that trans eQTLs explain about 10% of the genetic variance ($V_g$) on average. This is much less than the average cis contribution to $V_g$ (Fig 3A) and usually constitutes a minority of the overall $V_{g(trans)}$ for genes.

There are two reasons for lower diversity at trans eQTLs than cis eQTLs. First, trans eQTLs have a lower number of functionally distinct alleles per locus (median of 2 instead of 3; Fig 3C). Second, for a given allele number, the average heterozygosity is lower at trans than cis eQTLs because the latter exhibit a more even distribution of alleles (Fig 4). At two allele loci, a trans eQTL is more likely to be 1:9 than 5:5 for allele counts, while the reverse is true for cis eQTLs. With three alleles, there are a greater number of configurations, but cis eQTLs are over-represented in high heterozygosity categories (right side of Fig 4B) and trans eQTLs in

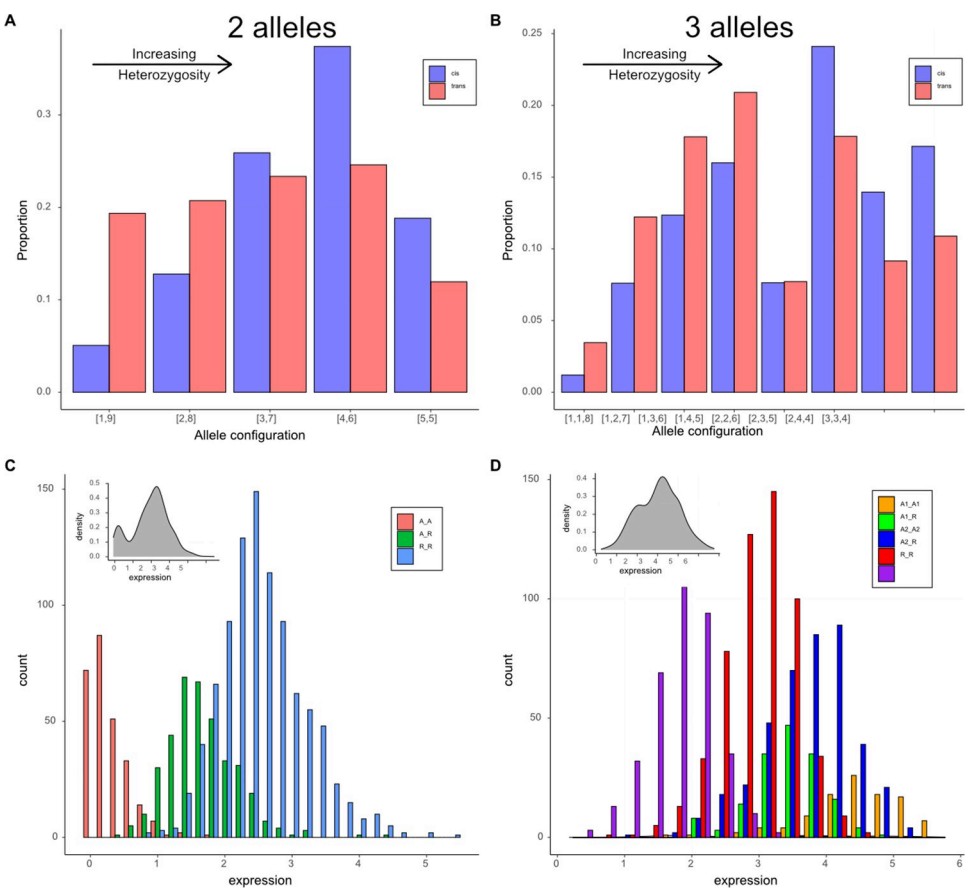

**Fig 4.** Panels A and B: The distribution of allele configurations for cis (blue) and trans (red) eQTLs for loci with (A) two alleles or (B) three alleles. The bracketed numbers refer to (A) the counts of the minor and major alleles or (B) the counts of minor, intermediate and major alleles. The allelic configurations are ordered left to right according to increasing heterozygosity (H). Panels C and D: Examples of the distribution of expression levels (Box-Cox transformed values used in the Combined analysis) per cis eQTL genotype (colored bars) or the overall distribution (inset for each panel). Panel C is a case where the cis eQTL has two functionally distinct alleles (gene = MgIM767.04G000700.v1.1) where five of the P lines carry the allele of IM767 (R) while the other four carry an alternative (A). Panel D is an example with three functional alleles (gene = MgIM767.10G016500.v1.1) where all P lines differ from 767 with two carrying allele A1 and the other seven carrying allele A2.

the lower categories (left side). The differences in the distributions of cis and trans are highly significant for both the two allele ($X^2$ = 505, df = 4, p < $10^{-16}$) and three allele ($X^2$ = 274, df = 7, p < $10^{-16}$) loci. Comparisons beyond three alleles are not possible owing to absence of trans eQTLs with high allele counts.

The partitioning of variation at eQTLs into functionally distinct alleles also illustrates an interesting aspect of the overall expression distribution of genes. Hsieh et al [34] noted that eQTL with major effects could generate a multi-modal distribution for expression–the distinct modes corresponding to the means of different genotypes. In this experiment, we do observe multi-modal distributions, but usually only when there is a large effect eQTL with only two alleles segregating. This is illustrated by a comparison of two major cis eQTLs, with either two (Fig 4C) or three (Fig 4D) distinct alleles. Inspecting the overall distributions (insets in figures), we see a clear bimodal distribution in the first case, but not the second. In either case, if we subdivide plants according to cis eQTL genotype, the underlying distributions reveal the cause of the difference. There is a relatively simple unimodal distribution within each cis eQTL genotype for both genes. However, they separate more clearly in the two-allele case simply because fewer genotypes (and thus fewer distributions) span the range of expression variation. Genotypes with intermediate means (frequently heterozygotes) fill the "valleys" in the overall distribution particularly with three or more alleles segregate at an eQTL.

The average Heterozygosity of trans eQTLs (H = 0.41) is only about two thirds that of cis eQTLs owing to the differences in allele number/evenness. This is a large difference, but less than suggested by literal extrapolation from the Cross-specific analysis where trans eQTLs typically showed in only one family. In the Combined analysis, the "typical" trans eQTL has the minor allele present in two of ten lines. We obtain only one significant test in the Cross-specific analysis owing to sampling error and limited power (the Beavis effect [35]). In fact, there are a small number of potentially important cases where the trans eQTL has high allelic diversity. For example, MgIM767.11G072100.v1.1 (a MADS box transcription factor) is affected by a QTL about 17.5 mb into chromosome 4 that segregates within four of nine families.

Considering the results from the perspective of the phenotype, we find that many genes are affected by multiple trans eQTLs, both within and between families. Across all genes measured for expression, the number of trans eQTLs ranged from 0 to 12. There is a strong positive relationship between the "trans heritability" ($V_{g(trans)}$ as a proportion of the total variance in expression) and the number of significant trans eQTLs identified for that gene in the Cross-specific analysis (Fig 5A). For a given number of trans eQTLs affecting a trait, $V_{g(trans)}$ increases with the amount of genetic variance generated by these loci ($F_{1, 6808}$ = 58.4, p < $10^{-13}$). In other words, the amount of variation contributed by mapped trans eQTLs is a strong predictor of the estimated $V_{g(trans)}$ of a gene (which is the total contribution of trans eQTLs, mapped or not). However, even in this subset of genes where we identified at least one trans eQTL, the majority of $V_{g(trans)}$ remains unexplained.

Allelic dominance at trans eQTLs is both more frequent and more severe than at cis eQTLs. In the Cross-specific analysis, 60% of trans eQTLs are significant for dominance and abs($d$) < abs($a$) in only 51% of cases (Fig 3D). This does not imply extensive over/under dominance because loci with complete dominance will produce point estimates with abs($d$) > abs($a$) about half the time owing to estimation error. We can test for over/under dominance by comparing the likelihood of the data with $d$ unconstrained to the likelihood under complete dominance of either allele. For cis eQTLs, this test provides only one gene with a compelling case for over/under dominance: MgIM767.14G274100.v1.1, a pectin acetylesterase protein, is highly significant for overdominance in three families and marginally significant in a fourth. There are more trans eQTLs suggesting over/under dominance, but none rise to genomewide significance.

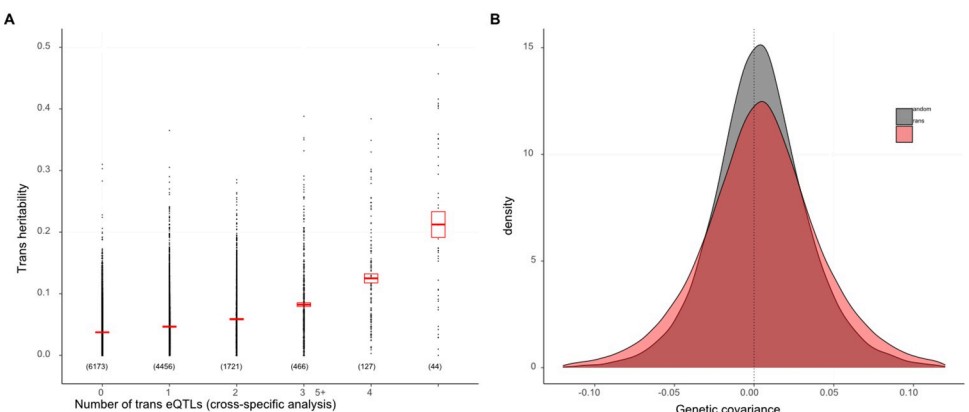

**Fig 5.** (A) Trans heritability, the fraction of the variance in expression explained by $V_{g(trans)}$, is predicted by the number of trans eQTLs affecting a gene. Number in parentheses is the count of genes in each category and the rectangle is the 95% CI on the mean. (B) The frequency distributions (density) for genetic covariance between genes affected by the same trans eQTL and all gene pairs.

## The contribution of trans eQTLs to genetic covariances

The additive genetic covariance between any two genes ($C_G$) can be estimated simply by applying the Combined model analysis to the sum of expression at the two genes and then subtracting estimates obtained from the fits to each gene alone (see Methods D). To characterize the genomic distribution, we randomly paired each expressed gene with 10 other genes and estimated $G_G$, and the environmental covariance ($C_E$), for each of these 64,930 trait pairs. $G_G$ and $C_E$ each exhibit distributions with a roughly equal mixture of positive and negative values (S7 Fig). With expression levels standardized to unit variance, we can use squared covariances to measure the magnitude of genetic and environmental associations. For $C_G^2$, the genomewide mean is 0.000983 (SE = 0.000007), while the mean for $C_E^2$ is 0.010545 (SE = 0.000081). These low values reflect the fact that most genes are uncorrelated with the bulk of the transcriptome (thousands of pairwise comparisons), even if strongly correlated with a subset of other genes (tens to hundreds of comparisons).

We tested the effect of trans eQTLs on co-expression by identifying all pairs of genes affected by the same trans eQTL. Over 305,809 gene pairs, the mean $C_G^2$ is 0.001639 (SE = 0.000007), which is the 67% greater than the genomewide average (t = 65.6, p < 10⁻⁹; Fig 5B). This confirms the prediction that trans eQTLs contribute to co-expression. While most gene pairs were specific to one cross, there were 621 pairs mapped in two families and 4 pairs mapped in three families. The magnitude of the genetic covariance (mean $C_G^2$) is much higher in these multi-family gene pairs ($F_{2,305806}$ = 2848, p < 10⁻⁹; S8 Fig). This is also expected given that intermediate allele frequency polymorphisms should generate more covariation than rare allele polymorphisms. An unexpected result is that $C_E^2$ is also inflated in trans eQTL pairs where the mean of 0.014266 (SE = 0.000053) is 35% greater than the genomewide average (t = 38.4, p < 10⁻⁹).

The standard approach to eQTL mapping is to progress gene by gene, predicting expression of each from genotype (as we did here, Figs 2–4). The obvious extension for correlations is to consider genotype effect on trait pairs. This is not a standard analysis, perhaps because the number of distinct gene pairs is very large. Instead, researchers typically apply methods such as principal components analysis (PCA) or network analysis [36] or sparse factor analysis [37] to compress correlated expression patterns into a tractable number of aggregate traits (PCs or factors or modules). For the present dataset, we applied PCA to the transcriptome and tested

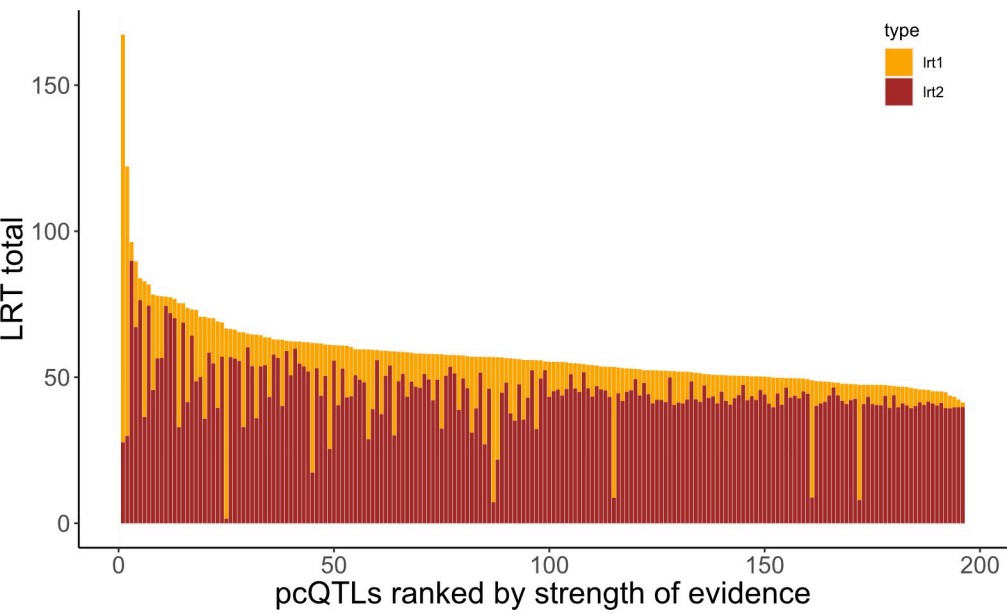

**Fig 6. The strength of evidence (LRT total) is reported for each of the 196 pcQTLs with test p-values $< 10^{-5}$.** Total LRT is the sum of the additive test $LRT_1$ (yellow) and the dominance test $LRT_2$ (red).

for genetic effects on the resulting principal component scores. PCA combines the expression values from correlated genes to define PCs that are linear combinations of all expressed genes. After defining PCs, we applied our Combined model analysis to each, treating them as quantitative traits. We determined the genetic and environmental variance in each PC and mapped loci affecting these composite traits (pcQTLs).

We found that the first 50 principal components explain nearly half of the total variance in expression of the full set of 12,987 genes. Importantly, the average heritability of PC scores is considerably higher than for individual genes (the mean is 0.25 for the first 50 PC and 0.34 for the next 50; S7 Table). We mapped 196 pcQTLs that yield test p-values less than $10^{-5}$ (Fig 6 and S8 Table). These co-localize with the major trans eQTL hotspots. Of the 35 hotspots, 24 have one or more pcQTLs within 400kb (which is +/- 2 centimorgans on average in the $F_2$ populations). Methods G provides a more quantitative contrast by considering the loadings of each PC on individual genes. This analysis indicates that the pcQTLs are absorbing both cis and trans eQTLs to some extent (Methods G). It is also noteworthy that the dominance test is usually more significant than the additive test on pcQTLs (Fig 6), which mirrors the prevalence of dominance for individual trans eQTLs (Fig 3D).

## Discussion

On average, the cis regulatory region of genes explains about two thirds of the genetic variance in expression of genes within leaf tissue of *Mimulus guttatus* (Fig 3A). This is an unexpectedly high proportion. Reviewing studies from a number of species, Liu *et al* [11] report that $V_{g(cis)}$ is typically about one third of the genetic variance, half our estimate. It is difficult to know if the Mimulus estimate is atypically high, because while large $V_{g(cis)}$ values are routinely obtained in eQTL mapping experiments (e.g. [31,33]), researchers usually only report estimates for genes with a significant cis eQTL. Our estimate and the summary by Liu *et al* [11] average over all measured genes. Methodological issues aside, a basic feature of *M. guttatus* may be relevant to its high cis regulation. This species exhibits remarkably high gene sequence

and insertion/deletion (indel) variation, even within local populations [38]. In fact, it is difficult to reliably map Illumina sequencing reads to intergenic regions of the *M. guttatus* reference genome where indel variation is very high [39]. Such variation could affect gene expression insofar as binding of regulatory elements to the DNA surrounding genes affects expression.

Speaking to the role of sequence variation, we find that genes with elevated indel variation upstream of the gene start codon and/or high nucleotide variation within the gene body have higher $V_{g(cis)}$ than do genes with lower variation (S5 Table). These patterns have at least two non-mutually exclusive explanations. First, genes with high variation *in general* might be more likely to exhibit differences in the regions that are directly relevant to gene regulation [40]. This is plausible but difficult to evaluate given that we cannot yet bioinformatically identify regulatory sequences (promoters, enhancers, etc.) in *M. guttatus*, but sequence variation in promoter regions has been correlated with the magnitude of cis eQTL effects in *Arabidopsis* [41]. Second, the level of sequence variation around a gene could be indicative of the history of natural selection at a locus. Genes under stronger purifying selection, or those that have recently experienced a selective sweep are expected to exhibit lower sequence variation. Such genes might also tend to exhibit lower standing variation in cis regulation. In corn, rare variants within cis regions are associated with 'dysregulation' of gene expression [42], although this is not apparent in *Mimulus* [43].

## Allelic heterogeneity and the allele frequency spectrum at eQTLs

Brown and Kelly [43] recently published a genomewide association study of gene expression variation (hereafter called the eGWAS) within this same IM population of *M. guttatus*. They examined a different plant tissue (flower buds instead of leaves), used a different experimental design (homozygous lines instead of lines intercrossed to produce $F_2$ individuals), and employed a different allocation effort (the eGWAS scored 151 lines with few individuals per line, while here we have 10 lines with high replication of segregating variation between lines). Despite these differences, the current experiment amplifies a key conclusion of the eGWAS: There is a striking difference in the allele frequency spectrum between cis and trans acting variants. "Cis-SNPs" have intermediate frequencies relative to the overall genomic distribution while "trans-SNPs" exhibit a rare-alleles model. The former is consistent with balancing selection on cis eQTLs while the latter suggests purifying selection on trans eQTLs [44,45].

The high allelic heterogeneity documented in the present experiment suggests that the eGWAS may have actually underestimated the level of variation at cis loci. The eGWAS tested biallelic SNPs rather than allelic series of haplotypes. The maximum possible heterozygosity (H) with two alleles is 0.5, which is much below the average H for our cis eQTLs which usually segregate 3–4 alleles per gene (Fig 3C). The regulatory regions of our founding lines are haplotypes that differ in both SNPs and indels at many positions. Closely linked variants exhibit linkage disequilibrium in the IM population (see S2 Table in [39]) aggregating mutations at distinct positions into functionally distinct alleles. This is an emerging empirical trend: Multi-parental mapping populations in both plants and animals find that QTLs are best described as allelic series and not binary alternatives [33,46].

We find that trans eQTLs have lower diversity than cis eQTLs, which corroborates the cis/trans difference in allele frequency discovered in the eGWAS. In truth, the current experiment overestimates the amount of variance generated by individual trans eQTLs because we only estimated the variance contribution of trans eQTLs that emerged as significant in the Cross-specific analysis. This is a simple manifestation of the Beavis effect [35], which does not apply to our cis eQTL because we could include all cis-loci in our estimation of effects (loci do not

have to be discovered as significant to be included). Even with this inflation of importance, our mapped trans eQTLs explain only about 10% of the genetic variance in their affected genes on average. Most of $V_{g(trans)}$ remains unexplained and thus represents the aggregate contribution of loci with effects below our detection limit.

Ascertainment also implies that we are overestimating heterozygosity at trans eQTLs. The eGWAS discovered many trans SNPs where the minor allele segregates at about 5% in the IM population (see Fig 2 of [43]). In the present design, we lose about half such loci just because the minor allele is not sampled into any of the ten parental lines. When such an allele is captured, its frequency in the experiment (at least 10%) is twice that in nature, which effectively doubles the heterozygosity estimate. Our estimation procedure is unbiased in the usual statistical sense (Methods F): Averaging over all trans-acting loci, underestimates (loci where we fail to sample the rare allele) will cancel overestimates (loci where we do sample the allele) yielding the true heterozygosity *on average*. However, since we always focus on the significant tests after the experiment is completed, an inflated heterozygosity estimate is inevitable.

### The multiparental mapping design enables the discovery of trans hotspots and the cis/trans difference for genetic dominance

A major advantage of multiparental mapping is that it can give a much better examination of rare alleles than GWAS [47]. GWAS typically have low power for rare alleles, alleles carried by few individuals in the experiment. As noted above, most rare alleles are not sampled into multiparental designs, but for those that are, there is high replication in measured individuals. The previous eGWAS [43] found many trans eQTLs but no hotspots. It is possible that the difference from the present study, where hotspots were evident in each cross, is biological (e.g. the bud transcriptome has a different architecture than the leaf), but a statistical explanation is more plausible. Because trans-acting alleles tend towards rarity, the minor allele was usually present in only 5–15 plants of the eGWAS. In this situation, a locus affecting many genes will yield genomewide significant tests on a minority of its targets just due to limited power. Most of the rare alleles present in the eGWAS were not sampled into the parental lines of this study. However, those included are likely carried by over 100 plants which enables reliable detection of trans effects.

To accurately estimate dominance at a QTL, we require substantial representation not just of alleles but also of diploid genotypes. Most multiparental mapping populations consist of inbred lines, which allows high replication of known genotypes. The replicated $F_2$ design involves crosses among lines, not only to produce mosaics of the parental genomes (as is true of Recombinant Inbred lines e.g. [46]), but also to generate QTL heterozygotes. Owing to this feature, we find a striking difference in dominance between cis eQTLs, which tend toward additivity, and trans eQTLs that typically exhibit dominance (Fig 3D). The molecular biology of gene expression predisposes cis eQTLs to additivity. If each cis allele contributes independently by affecting only the linked gene copy, then additivity of overall expression results from a simple dosage effect. This logic does not apply to trans acting loci, but it is not clear why they should be so strongly skewed towards strong dominance at most loci. About 60% of trans eQTLs yield an absolute value for d/a that is greater than 0.75, which means that the heterozygote is closer to one of the homozygotes than to the additive midpoint.

While certainly more nearly additive than trans eQTLs, we can reject pure additivity of gene action at over 20% of cis eQTLs. At these loci, heterozygote expression is nearly always intermediate (Fig 3D), but often closer to one homozygote than the other. A subtle deviation from the midpoint is expected given that we impose a non-linear transformation on read counts prior to estimating allelic effects. For example, the $\log_2$ transform, which is fashionable

in gene expression studies, will tend to pull the heterozygote expression slightly towards the homozygote with higher expression at an additive locus. More substantial deviations suggest a feedback mechanism where expression of one allele is affected by the other. Autoregulation, which is well established in plants [15], provides one such mechanism. For example, transcription factors can increase or decrease their own transcription level by binding their own promoter region. However, sequence differences in either the protein or the regulatory region could direct the feedback (enhancement or suppression) more to one allele than the other.

## Trans eQTLs and genetic correlations in gene expression

The quantitative genetic summary of gene co-expression is the "G matrix" [48]. Each of the n expressed genes is represented by a row and column in an n x n dimensional matrix with the additive genetic variances in expression on the diagonal. The additive genetic covariance of two genes is reported in the off-diagonal matrix elements corresponding to these rows and columns. The G matrix for the transcriptome is expected to be "sparse" relative to that for morphological traits. Morphological traits that emerge from common developmental processes routinely exhibit moderate to high correlations. In contrast, while individual genes may interact strongly within "expression modules" [49], we expect most interactions to be weak or at least diffuse. Consistent with this expectation, we find that genes typically have a low additive genetic covariance ($C_G$) with most other genes (grey in Fig 5B). Our experiment shows that cis eQTLs are the primary determinant of the G matrix diagonal (Fig 3A) while mapped trans eQTLs contribute incrementally to genetic correlations in expression. The latter effect is subtle for individual loci (Fig 5B), which may reflect the fact that our mapped trans eQTLs explain a minority of the genetic variation generated by trans acting loci.

Given the high dimensionality of our G matrix (there are over 84 million distinct off diagonal terms), we applied principal components analysis (PCA) to the transcriptome and then mapped QTLs for the PC scores (pcQTLs). For a plant, a PC score is a linear combination of the standardized expression levels at each gene. The weights (loadings) differ among the PCs, but in our case, each PC is strongly influenced by hundreds to a few thousand genes (S9 Table). Thus, pcQTLs likely affect many genes, although the effects on individual genes may be modest and below our detection limit for individual trans eQTLs. That said, there is a clear indication that pcQTLs "capture" some of the effects of our mapped eQTLs, both cis and trans. This is simply because the PC affected by a pcQTL tends to have higher loadings on genes with genomically proximal eQTLs (Methods G). The association with trans eQTLs is stronger than with cis. Also, both pcQTLs and trans eQTLs show a much stronger signal of genetic dominance than do cis eQTLs (Figs 3D and 6).

PCA is a classic tool in quantitative genetics. It is applied directly to correlated traits to obtain uncorrelated predictors of fitness [50] and also to characterize the structure of the G matrix [8, 51, 52]. PCA is also used in RNAseq experiments, often for data visualization but sometimes as a data cleaning tool to remove "confounding factors." If an environmental variable (say temperature) influences the expression of many genes, the failure to control for this variable can reduce the power to detect treatment effects. If the leading principal components 'absorb' the effects of unmeasured variables, the inclusion of PC scores as covariates can remove noise and improve power. Our results are cautionary with respect to this approach if genotype is the treatment. Genotypic differences generate a correlated response across genes. Pleiotropy can thus be (partly) responsible for co-expression patterns that determine principal components. In this experiment, we found that PC scores actually have a higher genetic determination than individual genes on average (S7 Table). To statistically remove PCs before analyzing individual genes can thus eliminate signal (trans eQTLs) as well as noise.

## Concluding remarks

Evolutionary inferences from genetic experiments always depend on sampling, on where genotypes come from. Multiparental mapping populations are typically created from world-wide collections [46,47,53,54], a strategy designed to maximize genetic diversity. Most eQTL experiments that have been done in plants are based on broad geographic samples or on crosses between genotypes chosen specifically because they exhibit interesting (or agriculturally important) phenotypic differences. In these experiments, the frequency of alternative alleles within the mapping population will be determined by the chosen parents, as will anything that depends on these frequencies such as QTL variances. If parents are sampled across natural habitats, then genetic variants responsible for local adaptation will segregate in the mapping population. In contrast, experiments estimating quantitative genetic (co)variation are typically based on a random sampling of genotypes from a specific population. This ensures that allele frequencies in the mapping population are representative of, and informative about, the ancestral population. There are fewer studies of this kind, but recent work in both *Capsella grandiflora* [45] and *Populus tremula* [55] have estimated the contribution of individual loci to the standing genetic variance in gene expression.

In this experiment, we sampled parental genotypes from one natural population (Iron Mountain) with the purpose of estimating features of that population. This is very large population that reproduces mainly by outcrossing (at a rate of over 90% in most years [56,57]). Moreover, because of very high inbreeding depression [58], adult plants are almost entirely outbred at Iron Mountain. We founded this experiment from a collection of inbred lines made from randomly sampled Iron Mountain plants and previous sequencing confirms that these line population is representative of the ancestral population in terms of allele frequencies [25,39,43]. These allele frequencies thus reflect the balance of evolutionary processes (selection, migration, drift) at the Iron Mountain location. The high variation at cis eQTLs suggests that selection is maintaining variation at this local scale [43]. Field studies at Iron Mountain directly measuring selection on genetic variants [25,59], as well as longer term studies measuring temporal changes in allele frequency [39], suggest that antagonistic pleiotropy and temporally fluctuating selection are both acting as selective agents that maintain polymorphism.

Trans eQTLs have lower allelic diversity than cis eQTLs and a greater contribution of uncommon alleles. However, the aggregate of evidence from the current experiment and the previous eGWAS suggests that these relatively "minor" alleles segregate in the 1–10% range within Iron Mountain. This is considerably higher than the expected frequency of unconditionally deleterious alleles, which are likely to be less than 1% in a large population. The scale of sampling is a key consideration here. An allele that is uncommon within Iron Mountain, perhaps because it is usually disadvantageous under local environmental conditions, may be predominant in other populations. As in most widely distributed species, local adaptation is very common in *M. guttatus* (e.g. [60,61]). If trans eQTLs are important to local adaptation in *M. guttatus*, we predict that the allele frequency spectrum for trans eQTLs will shift when we apply the same experimental design to a species-wide sample of parental genotypes.

The characterization of gene expression in terms of genetic variances and covariances is necessary to predict the response to natural selection. Field experiments have demonstrated rapid evolution of gene expression in response to selection [62–64]. From one generation to the next, the change in mean expression levels under selection can be predicted from the current G matrix without any information on the genetic architecture of expression variation [65,66]. However, the rate that G matrix elements change is dependent on how eQTLs combine to determine genetic (co)variances. Our finding that a major locus, the cis eQTL, explains much of expression variation suggests that the G matrix will be malleable on ecological time

scales. Shifts in allele frequencies at major QTLs rapidly change genetic variances and covariances [6,67]. The usual view is that selection eliminates genetic variation, which should occur rapidly if fixation at one gene eliminates much of the variation. However, with the temporal fluctuations evident at Iron Mountain, variation can persist even with strong selection [68,69].

The finding of high allelic diversity at eQTLs further complicates G matrix dynamics, particularly when considering genetic covariances. With two alleles and additive gene action, we can describe a locus as either positive or negative with respect to the covariance of two affected traits. If the first allele increases expression at two genes, the alternative necessarily has a negative effect on both (because effects are defined by contrast between alleles). With multiple alleles, this is no longer assured. With four alleles, all directions for pleiotropic effects could be evident (positive/positive, negative/negative, positive/negative, and negative/positive). The extent to which allelic heterogeneity generates complex pleiotropy is currently unclear, making it an important target for future experimental work.

## Methods

Each of the sections below reference computer programs for analysis. All software developed by others, including standard bioinformatic tools such as Salmon [70] and Gemma [23], are reported as used. Most of our analyses were completed using custom programs written in Python (v3.7). These programs are provided in S1 File along with "Key_to_programs.docx" which describes their use.

### A. Study system, experimental protocols, RNA and DNA extractions, RNAseq library preparation, and sequencing

As parents, we used inbred lines of the yellow monkeyflower, *Mimulus guttatus* (syn *Erythranthe guttata*, Phrymaceae) extracted from the Iron Mountain (IM) population in the Cascade Mountains of Oregon (44.402217N, 122.153317W; [56]). This population is predominantly outcrossing with little internal population structure [71]. In 2018, Troth et al. [25] sequenced whole genomes of 187 IM inbred lines. After selecting one of these lines (767) as the "reference" (which is common to all crosses), we sampled the nine alternative lines subject to the condition that they were fully unrelated to 767 and each other. Relatedness was based on genomewide pairwise nucleotide diversity ($\pi$) from the Illumina sequencing of these lines [25]. The equidistance among the lines is confirmed by the subsequent *de novo* assembly of each line from long read sequencing data (described below, see S1 Fig).

We crossed each alternative line to 767 with the latter used as the pollen donor (Fig 1A). We grew a single F1 plant from each cross and self-fertilized this individual extensively to produce >1,000 seeds. We grew the $F_2$ plants along with members of each parental line in four temporally overlapping cohorts using standard greenhouse conditions [43], about 500 plants per cohort. Each family was grown in two cohorts and the IM767 line plants were grown in all four cohorts. Daylength was kept at 16 hours (supplemental lighting on at 6am, off at 10pm) throughout the experiment. We collected whole leaves from the $2^{nd}$ leaf pair as soon as the third leaves were >1cm long (Fig 1B) and immediately flash froze the tissue in liquid $N_2$. All leaves were collected between 10am and noon to control circadian rhythm effects on expression. RNA was extracted after disrupting the frozen leaves with metal beads using a bead beater in RLT and $\beta$ mercaptoethanol solution using a Qiagen RNeasy plant mini kit (Qiagen) according to manufacturer's instructions (without the optional DNase step). All samples were eluted in 60 $\mu$l RNase-free $H_2O$.

We made RNAseq libraries using QuantSeq 3' mRNA-seq Library prep kits (Lexogen) at quarter volumes. We used four i5 primers (Lexogen), which along with the 96 i7 primers of

the kit, allow barcoding of 384 samples per sequencing run. Each batch of 96 samples from one run of the protocol was pooled in equal volumes and checked for fragment size distribution using an Agilent TapeStation (Agilent, Santa Clara, CA, USA) and quantified using Qubit (Thermo Scientific) at the KU Center for Genomics. Four such batches, each with different i5 primers, were pooled equimolarly for a sequencing run. Sequencing was performed at the KU Center for Genomics on a NextSeq 2000 to obtain 75bp single end reads. For samples with low yield in sequencing, we remade libraries from the original RNA extraction and sequenced the remade libraries.

## B. *de novo* assembly of parental genomes and annotation

From plants of each parental line, we extracted DNA using a modified PacBio protocol for high molecular weight DNA extraction using 5 g leaf tissue as starting material. The full protocol is available as S2 File. After confirming high molecular weight using an Agilent TapeStation (Agilent, Santa Clara, CA, USA), we sent the extracted DNA to the University of Georgia where Sequel II CLR libraries were prepared and sequenced according to the manufacturer's instructions. We extracted fasta files from the PacBio raw data (SMRT Link XML) using the ccs and bam2fastq commands from smrtools v9.0.0.92188 (PacBio). We assembled genomes using canu 2.1.1 [72] with options genomeSize = 430m, correctedErrorRate = 0.035, utgOvlErrorRate = 0.065 trimReadsCoverage = 2 trimReadsOverlap = 500. The resulting assemblies were reduced to haploid assemblies using purge_dups [73], and were scored for quality using BUSCO v3.0.2 [74] and the embryophyta_odb9 dataset. For the data analysis of this study, we used the genome assemblies of 767 and 62 produced by the Joint Genome Institute ([24], used with permission). Our Pacbio/Canu assemblies were used for the other eight lines.

We used Liftoff [75] to transfer the annotation of the 767 assembly onto the other genomes. Given an annotation file (gff3) from each assembly, we identified orthologs of each 767 gene in each alternative build. Not all 767 genes were successfully located in the other assemblies and so we focused on the 12,987 genes discovered in all lines. These genes also passed the minimum average expression level described below. We extracted the sequence of these genes in each build to create a line specific transcriptome for read mapping. To score differences in the nucleotide sequences among our assemblies, we used Mummer 3.0 [76] and SVMU [77] as described in Program Set 1 of Key_to_programs (S1 File).

From the output files, we extracted all correctly aligned positions between genomes. We scored SNPs and indels (which were enumerated as gaps in 767 relative to the alternative genome and gaps in the alternative relative to 767) in each gene and in the 1kb upstream of the start codon and downstream of the gene end. For each interval, we calculated nucleotide diversity ($\pi$) as the fraction of aligned positions that differ in nucleotide and U = (total bp–aligned bp)/ total bp. If the two sequences are perfectly colinear (no indels) then total bp = aligned bp and 1 with 0. U increases towards 1 as the region fills with indels and unalignable sequence. This yields six statistics for each gene ($\pi$ and U for each of the three intervals), which we used as predictors of the LRT1 test statistic for a cis eQTL. We standardized each predictor (to mean zero and variance 1 across all genes) and then applied multiple linear regression using Minitab v19 to produce S5 Table.

## C. Read mapping, genotype calling and scoring expression levels

We trimmed the RNAseq reads with Trimmomatic [78] and checked for contamination or mislabeling using custom python scripts (S1 File) that estimated the relatedness of all among samples including the parental lines. After eliminating dubious samples (low sequencing depth or questionable family assignment), we retained data from 1588 plants for subsequent

analysis. We used Salmon v1.10.0 [70] to quantify gene expression. To remove bias caused by preferential mapping of alleles, we mapped the F2 individuals from each family to a composite genome including the (haploid) genome and transcriptome of each parental line. We excluded any gene that displayed aberrant mapping of reads from the parent line plants. Specifically, we required that reads from the inbred line plants (which are homozygous for a known parental allele at all loci) map specifically to the allele from that line (in which case the marker is informative for genotype as well as transcript level analysis) or that the line alleles map equally well to each allele. Only genes in the former category were amenable to allele specific expression analyses.

Within each cross (family), genotyping was based on the RNAseq data from the subset of genes where reads reliably map to each parental allele (identifying their origin). We used the count of reads to each parental allele to make a putative call within each marker locus (RR, RA, AA). These calls are error prone owing to low read counts at many loci (lowly expressed genes) and allele specific expression (which makes heterozygotes resemble the homozygotes associated with higher expression). Thus, we treat these putative calls as the "emitted states" with the true genotype treated as the latent states of a Hidden Markov Model (HMM). The HMM estimates the genotype of each F2 plant for each chromosome and is implemented using a series of python programs revised from the GOOGA pipeline [27]. The model estimates marker-specific genotyping error rates (which determine emission probabilities) and the recombination rates between all adjacent markers (which determine the transmission probabilities of the Markov Chain).

Given maximum likelihood estimates for all parameters, we extract the genetic map for each cross and the posterior genotype probabilities at each marker. The locations of 33,302 recombination events across 1,373 $F_2$ plants is reported in S10 Table. The resulting genotype matrices are nearly complete given that the posterior probabilities for the most likely genotype at each marker are (almost) always greater than 0.99. To produce a genotype matrix with calls at each expressed gene, we interpolated calls from the scored markers that were immediately upstream and downstream of any gene not included in the HMM estimation (these are all genes without informative markers for parental assignment of reads). When adjacent markers differed owing to a recombination event, intermediate genes were scored as unknown genotype. Finally, we calculated the relatedness matrix using pairwise comparisons among all 1,588 plants. The coefficient of coancestry at each marker is determined simply by the extent that the two individuals share alleles from the same parent line given our assumption that parental lines are unrelated. The overall relatedness between two plants (twice the coefficient of coancestry) is just an average across all loci. These programs to infer genotypes and relatedness (as well as those used for other aspects of the data analysis) are contained in S1 File with an outline describing the sequence that programs are executed and the inputs/outputs for each step.

To obtain the phenotypes (the total expression of each plant at a gene), we summed the reads mapped to each allele of a gene within each plant. Lowly expressed genes, those with a mean expression less than 0.5 count per million, were not considered for further analysis. For the Cross-specific analysis, we estimated the mean-variance relationship using Voom [79]. Voom also generated a weight for each observation considering the growth cohort and Cross as factors. First, a DGEList object was generated in edgeR [80] from the raw counts, which included information on library size per individual using the calcNormFactors command. Then the DGEList object was Voom transformed given a matrix of cohort + group. The resulting normalized counts and weights were exported for the Cross-specific analysis. The Combined analysis used the standardized counts per million for each gene of each plant directly. A Box-Cox transform was then applied, gene by gene, and finally standardized the transformed counts to unit variance.

For alleles specific expression analysis, we identified all genes in each cross where the total expression could be partitioned into reads contributed by each parental allele in heterozygotes. At such genes, we assembled a vector of count pairs (reads from the reference, reads from the alternative line) for all plant heterozygous at the cis locus. We treat these counts as samples from a beta-binomial distribution. We determine the maximum likelihood across all plant under the null model enforcing $\alpha = \beta$ (which implies no allele specific expression on average, equal expression of both parental alleles) to the more general alternative model $\alpha \neq \beta$ (either parental allele can be overrepresented). The beta-binomial is superior to the usual binomial model for counts because it allows over-dispersion. However, we find that our MLE for the frequency of the alternative allele in heterozygote RNA (y-axis of Fig 3B) from the beta-binomial is usually very close to the "naïve estimate", which is the simple average of A/(R+A) across all heterozygous plants.

## D. Testing procedures

For the Cross-specific analysis we used rQTL v 1.60 [81] to detect QTLs for the normalized expression of each gene using the scanone function with cohort as a covariate and taking the weights into account. The analysis was run separately on each family, and we extracted the LOD score, additive and dominance effects, and their standard errors for each marker. P-values were obtained from permutations specific to each gene using the scanone command of rqtl and the marker regression method. For all trans eQTLs, the location of the eQTL is reported as the location of the LOD peak. However, for LOD peaks near the expressed gene, we tested the marker closest to that gene (oftentimes the marker is the gene itself). The marker was considered proximal if within the LOD confidence interval from rQTL and also less than 1mb distant from the gene start site. We called a cis eQTL only if the LOD at the gene location was significant (exceeded the genomewide threshold). If significant, we extracted the estimates for effect (a and d) from this location.

For the Combined analysis, we fit a linear mixed model using maximum likelihood to the entire dataset for each gene, considering three different models in sequence. The calculations were performed using Gemma [23] as described in Program Set 3 of the Key_to_programs (S1 File). The fixed effects in the "null model" are cohort and the random effect absorbs all genetic effects on expression (the relatedness matrix determining the (co)variance matrix). This Model 0 yields a log-likelihood ($LL_0$) and two variance estimates, Vg and Ve. Vg is the (whole genome) additive genetic variance in expression while Ve is the residual variance (environmental effects, measurement error, etc.). We next test for a cis eQTL by adding the genotype at this locus into the vector of fixed effects. We first consider purely additive gene action at the cis locus: Model 1 adds nine parameters, the effects of each alternative allele (specific to each line) that is crossed to 767. For all plants from cross z, the phenotype is incremented by $a_z$ for heterozygotes and by $2 a_z$ for homozygotes for the allele from parental line z. Model fitting yields the log-likelihood ($LL_1$), estimates for all nine $a_z$ values, and the variance components, Vg and Ve. In the fit of model 1, Vg is the genetic variance due to trans eQTL since any cis-locus effect has been absorbed into the fixed effects. The likelihood ratio statistic, $2(LL_1—LL_0)$, is compared to a chi-square distribution with 9 df to test for an effect of the cis-locus. Finally, Model 2 allows dominance at the cis eQTL with genotypic values of 0, $a_z+d_z$, and $2a_z$ for reference homozygotes (767), heterozygotes, and alternative homozygotes (line z), respectively (the rQTL model used in the Cross-specific analysis). Model 2 adds nine parameters, so the likelihood ratio test for dominance, $2(LL_2 –LL_1)$ is compared to a chi-square distribution with 9 degrees of freedom. We first applied models 0, 1, and 2 to each gene using the cis locus as the

genotype. Later, after identifying affected gene / trans eQTL pairs from the Cross-specific analysis, applied these models to each pair using the trans eQTL locus as the genotype.

From the ML model fits, we can estimate the genetic variance generated by the cis eQTL in two different ways. First, we can subtract the Vg from model 1 (which includes only trans effects) from the Vg of model 0 (which includes both cis and trans effects). This estimator, denoted $Vg(r^2)$, is similar to the 'variance explained by the QTL' method that has been applied to multi-parental mapping populations of Drosophila [33] and mouse [31]. A second estimator, Vg(a), is calculated from the estimated additive effects and their standard errors:

$$Vg(a) = 2(s_a^2 - \bar{se}_a^2)$$

where $s_a^2$ is the variance among the nine $a_z$ estimates and $\bar{se}_a^2$ is the average of squared standard errors on those estimates. This formula is the simple variance ($s^2$) among $a_z$ values minus the variance generated by estimation error. The $a_z$ estimates can be treated as unrelated because each is calculated from genotype/phenotype association within distinct families, The simulations described below indicate that both Vg(r2) and Vg(a) are (nearly) unbiased but Vg(a) has a lower mean square error. In other words, for this design Vg(a) is closer to the true variance on average. Both estimators perform substantially better than the HE regression approach [30], which also is nearly unbiased but has much larger mean square errors.

The alternative models (1 and 2) described above are fully unconstrained–each alternative line can be uniquely different from 767. This is necessary if variation is described by an 'infinite alleles' model [82], but over-parametrized if few alleles segregate at the cis locus. All the possible configurations, ranging from two to ten distinct alleles at a QTL, can be testing by first ranking the $a_z$ estimates from most negative to most positive and then considering all "splits" that subdivide these estimates into discrete bins [33]. We determined the maximum likelihood for all 511 distinct configurations for each gene. For a given number of partitions (e.g. two partitions equals three alleles), we selected the case with the highest likelihood. Starting with the 2-allele case, we accept increases in the allele number (rejecting the simpler model by accepting an additional parameter) only if twice the likelihood difference is greater than 3.84 (which is the p<0.05 threshold for a chi-square test with 1 df).

The linear mixed model was applied in two other analyses. To test cases of potential over/under dominance, we compared the likelihood of the data with $d$ unconstrained (Model 2) to the likelihood under complete dominance of either allele. This has the same number of parameters as Model 1 but with a different assignment of genotype effect to heterozygotes. Second, we applied Model 0 to the sum of expression (Z) between pairs of genes ($y_1$ and $y_2$): Z = $y_1$+$y_2$. This yields estimates for genetic and environmental variance in Z. For either genetic or environmental components, we note that $Var[Z] = Var[y_1] + Var[y_2] + 2Cov[y_1, y_2]$. Consequently, we can solve for the genetic ($C_G$) and environmental ($C_E$) covariances given the corresponding variances in Z and as well as the single trait estimates for $V_G$ and $V_E$. For all tests, we obtained a p-value across all 12,987 genes and then assign the False Discovery Rate (Q-values) using "p.adjust()" in R [83].

## E. Mapping QTLs for expression principal components

We created a matrix with standardized expression levels for all genes (mean zero and variance one) for all plants prior to invoking Principal component estimation programs. Using the R libraries MASS, dplyr, and data.table, we applied the prcomp function to the expression matrix and extracted the variance explained by each principal component, the loadings on all 1,588 PCs, and the PC scores for each plant for each principal component. We formatted each PC score list as a phenotype file for input to the Combined analysis pipeline. We first fit the linear

mixed model to each PC score without a QTL to estimate the genetic and environmental variance of that trait. Next, we added a QTL single to the model with the position adjust incrementally along each chromosome. Here we tested every other gene location (a step size $\ll$ 1 cm). We retained the model fits with the highest log-likelihood per chromosome as putative pcQTLs.

## F. Simulations to test estimation procedures

Our testing and estimation procedures were evaluated using simulations grounded in the design of our experiment. Specifically, we considered a range of scenarios with and without genetic variation in expression by simulating phenotypic (expression) data from the observed genotypes of our 1588 plants. Each simulation replicate starts with the selection of a random gene. Given this, we can distinguish the cis genotype for each plant (which will affect phenotype if we are simulating a cis eQTL) as well as the genomic background (which will affect any simulated trans effect). To each simulated dataset, we fit the Models 0 and 1 applied to the actual data, as well as several alternative approaches. For each ML fit, we applied two methods to estimate the variance generated by the cis eQTL, Vg[r2] and Vg[a] as defined above in the Methods. We also tested the Haseman–Elston (HE) regression [30] instead of ML to estimate $V_E$, $V_{g(cis)}$, and $V_{g(trans)}$.

The first set of simulations consider the null case where all variation is environmental ($V_E$ = 1, $V_{g(cis)}$ = $V_{g(trans)}$ = 0). Random normal deviates were generated using the normal() and multivariate_normal() functions from the NumPy library of python [84]. In this case, we find that the likelihood ratios tests very nearly follow the predicted null distribution (S9 Fig) but are slightly conservative–only 3.5% of tests of model 1 yield p < 0.05 instead of 5%. In terms of variance estimates, bias is minimal for both ML and HE regression, both with and without genetic variation in expression (S3 Table). This is noteworthy given that ML imposes feasibility constraints (e.g. $V_E$ cannot be negative) while HE regression does not. To simulate data with cis effects on expression, we sampled a unique additive genetic value for each line from a normal distribution given a specified value for $V_{g(cis)}$. We simulate a trans effect generated by many small effect loci by sampling a vector from the multinormal distribution. The covariance among plants is determined by $V_{g(trans)}$ and the relatedness matrix. The mean of estimates from both ML and HE regression are close to the true values regardless of whether model 0 or model 1 is fit to the data. In other words, all methods are approximately unbiased. However, ML is far more precise than HE regression when there is genetic variation in expression. The root mean square error (a measure of the magnitude of estimation error) is much smaller for both $V_{g(cis)}$ and $V_{g(trans)}$ (S3 Table). We find that Vg[a] is slightly but consistently more precise than Vg[r2] in estimating $V_{g(cis)}$. For this reason, we report Vg[a] from the ML fits to each gene.

While not included in S3 Table, we performed many additional simulations to consider additional data types and alternative analytical procedures. First, we considered the case where only two alleles segregate at the cis eQTL. For these simulations, we randomly sampled from a bi-allelic locus where the positive allele is at population frequency q and has a fixed additive effect a. Given q and a, $V_{g(cis)}$ = 2 q(1-q) $a^2$ [85]. We found that Vg[r2], Vg[a], and HE regression all yielded unbiased estimates for $V_{g(cis)}$ and the pattern for precision (Vg[a] better than Vg[r2] much better than HE) was unchanged from the infinite alleles model simulations in S3 Table. Next, for both ML and HE regression, we considered the "leave-one-out" option [86] for relatedness calculations. With this option, the trans genetic effect considers all chromosomes except the one that harbors the expressed gene and putative cis locus. When $V_{g(cis)}$ = $V_{g(trans)}$ = 0, estimation and hypothesis testing outcomes are essentially unchanged using leave-

one-out. Using leave-one-out in simulation that include either cis- or trans- genetic variation, the LRT values for model 1 are inflated. This is expected when there is a cis eQTL because the leave-one-out procedure is designed to increase power. Unfortunately, we also see that they are inflated in simulations where $V_{g(trans)} > 0$ but $V_{g(cis)} = 0$. This implies an elevated false positive rate. This occurs because the fixed effect parameters describing the cis eQTL can "absorb" the effect of trans eQTLs that are on the same chromosome, loci that are not included in the relationship matrix with leave-one-out. In our simulations, we assume that trans loci are distributed uniformly over the genome. For genes on large chromosomes (e.g. chromosome 14 in Mimulus), a substantial fraction of the trans-effect will emanate from genes on the same chromosome. We also find that the variance component estimates can be poorly behaved in the leave-one-out model fits applied to our experimental design. For these reasons, we used the overall relationship matrix in analysis of the actual data. This method might be underpowered in general, but that is not a major difficulty for the current study given that nearly all cis eQTLs were significant anyway (see Results).

## G. Estimating the overlap of pcQTLs with eQTLs

To assess overlap of pcQTLs with the eQTLs, we defined a genomic window around each pcQTL of +/- 2mb from the LOD peak. Noting the specific PC affected by the pcQTL, we determined the loadings of this PC onto each expressed gene (12,987 values). For testing against cis eQTLs, we asked if the loadings on genes within the window around the pcQTL were larger in magnitude than loadings on gene elsewhere in the genome. For a single pcQTL, we compared the means from two lists: the squared loadings on genes within the window (median of 263 per pcQTL) versus the genes outside the window (median of 12724 per pcQTL). We then performed a paired t-test on the differences across all pcQTLs. The loadings were larger within windows ($t_{195} = 2.26$, $p = 0.025$).

The test for overlap with trans eQTLs is similar in structure, a comparison of loadings within the pcQTL window versus those outside. However, here we surveyed the list of trans eQTL / affected gene pairs. For each such pair we noted whether the trans eQTL genomic location (the position of the causal locus) was within the pcQTL window or not. If so, the loading on the affected gene (which would generally reside elsewhere in the genome) would be added to the "within window" list. If the trans eQTL was located outside the pcQTL windows, the loading of the affected gene was added to the outside window list. Within window lists contained a median of 185 affected genes while the median count was 9874 for the outside list. As previously we distilled each list within each pcQTL into a mean of squared values and compared them across pcQTLs using a paired t-test. The loadings were larger on genes with trans eQTL located within windows ($t_{195} = 2.92$, $p = 0.004$).

## Supporting information

**S1 Table. A summary of features from our de novo assemblies based on PacBio sequencing.** N50 = the length of the shortest scaffold in the ranked list that covers at least 50% of the assembly.
(DOCX)

**S2 Table. The nucleotide diversity (π) and indel frequency (U) is reported for each gene in three intervals, within the gene body, the 1kb upstream of the gene, and the 1kb downstream of the gene.**
(XLSX)

**S3 Table. Simulations with cis alleles sampled from a normal distribution with Vg(cis) specified.** The color coding (green for r2, blue for [a], pink for HE) identifies the standard deviation among replicate simulations for each model for the non-zero genetic parameters in each simulation case. Since the estimators are (nearly) unbiased, the standard deviation estimates the root mean square error. For cis genetic effects, the error is smaller for blue than green and both are much smaller than pink. For trans genetic effects, the error is the same for green and blue (the estimators are the same) but again much smaller than for pink.
(XLSX)

**S4 Table. The results from testing the cis eQTL at each of 12,987 genes scored for expression.** "Number of alleles (cis eQTL)" and "Heterozygosity at cis eQTL" are derived from the allele partitioning method applied to each gene. LRT1 and LRT2 are the likelihood ratio statistics testing for additive and dominance effects, respectively. The p-value and q-value (FDR) is reported for each test. The last three columns report the variance component estimates for the LRT1 model fit: Ve, Vg_trans, and Vg_cis.
(XLSX)

**S5 Table. A regression of LRT1 values onto nucleotide diversity (p) and indel diversity (U) within three regions about each gene.** Each predictor was standardized to unit variance (z transform) to make the regression coefficients comparable.
(DOCX)

**S6 Table. Significant trans eQTL tests are distilled into 1,979 loci.** Each locus is given a unique name (first column) and then identified to cross and genomic location. The last two columns are the number and identity of affected genes.
(XLSX)

**S7 Table. The estimated environmental (Ve) and genetic (Vg) variance for each PC score is reported for the first 200 PCs. $h^2$ equals Vg/(Vg+Ve).**
(DOCX)

**S8 Table. For each of the 196 pcQTLs with $p < 10^{-5}$, we report the genomic location, affected PC, p-value, number of distinct functional alleles, heterozygosity, the additive, dominance, and total LRT tests (LRT1, LRT2, LRT_total).**
(XLSX)

**S9 Table. The loadings on standardized traits for the first 200 expression principal components.**
(XLSX)

**S10 Table. The locations are reported for 33,302 recombination events detected across 1,373 $F_2$ plants.** Each event is localized by differing genotypes (each with posterior probability > 0.99) at bracketing markers (left and right of the recombination breakpoint). The location of these flanking markers is reported as bp position in the IM767 reference genome.
(XLSX)

**S1 Fig. The fraction of nucleotide positions that differ (π) within genes is reported between each pair of lines.** We calculated π for each chromosome, and then averaged these 14 values to obtain the mean and standard error (the latter used to calculate the 95% CI).
(PDF)

**S2 Fig. The eQTL plot of all nine families (crosses).**
(PDF)

**S3 Fig. The rate of recombination within a chromosomal region affects average gene expression.** We divided all gene regions into quartiles (x-axis) and averaged the Log CPM across genes within each region.
(PDF)

**S4 Fig. The correlation of additive effect estimates between the Cross-specific (x-axis) and Combined (y-axis) analyses is 0.96.**
(PDF)

**S5 Fig. The strength of evidence for a cis eQTL (LRT1) is positively correlated with $V_{g(cis)}$.**
(PDF)

**S6 Fig.** Mean expression (measured as CPM on a log scale) is a strong positive predictor of test significance (left) and a strong negative effector of $V_E$ (right).
(PDF)

**S7 Fig.** $G_G$ (left) and $C_E$ (right) each exhibit distributions with a roughly equal mixture of positive and negative values.
(PDF)

**S8 Fig. The magnitude of the genetic covariance (mean $C_G^2$) increases with the number of families that segregate a trans eQTL.**
(PDF)

**S9 Fig. The distribution of LRT1 (the test for an additive eQTL) across 5000 simulations when there is no eQTL.** The actual mean of simulations (8.77) is slightly below the null distribution predicted value of 9 for a chi-square distribution with 9 df.
(PDF)

**S1 File. This tarball contains all of S1 File (python code and the instructions to run all programs).**
(GZ)

**S2 File. The protocol is reported for High Molecular Weight DNA extraction.**
(PDF)

## Acknowledgments

We thank Stuart Macdonald, Jae Choi for comments on an early draft of this paper. Sequencing was done by the KU Genome Sequencing Core which is supported by the National Institute of General Medical Sciences (NIGMS/NIH) under award number P30GM145499. We thank the Department of Energy Joint Genome Institute and collaborators Lila Fishman and John Willis for pre-publication access and use of the IM62 and IM767 genomes from the ongoing Mimulus pan-genome project. The work (proposal: 10.46936/10.25585/60001364) conducted by the U.S. Department of Energy Joint Genome Institute (https://ror.org/04xm1d337), a DOE Office of Science User Facility, is supported by the Office of Science of the U.S. Department of Energy operated under Contract No. DE-AC02-05CH11231.

## Author Contributions

**Conceptualization:** Paris Veltsos, John K. Kelly.

**Data curation:** Paris Veltsos.

**Formal analysis:** Paris Veltsos, John K. Kelly.

**Funding acquisition:** Paris Veltsos, John K. Kelly.

**Investigation:** Paris Veltsos, John K. Kelly.

**Methodology:** Paris Veltsos, John K. Kelly.

**Project administration:** John K. Kelly.

**Resources:** John K. Kelly.

**Software:** Paris Veltsos, John K. Kelly.

**Supervision:** John K. Kelly.

**Validation:** Paris Veltsos, John K. Kelly.

**Visualization:** Paris Veltsos.

**Writing – original draft:** John K. Kelly.

**Writing – review & editing:** Paris Veltsos, John K. Kelly.

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
