## [Decision Letter · Decision Letter 0]

25 Jan 2024

Dear Dr Kelly,

Thank you very much for submitting your Research Article entitled 'The quantitative genetics of gene expression in Mimulus guttatus' to PLOS Genetics.

The manuscript was fully evaluated at the editorial level and by independent peer reviewers. The reviewers appreciated the attention to an important problem, but raised some substantial concerns about the current manuscript. Based on the reviews, we will not be able to accept this version of the manuscript, but we would be willing to review a much-revised version. We cannot, of course, promise publication at that time.

If you decide to revise the manuscript for further consideration at PLOS Genetics, please aim to resubmit within the next 60 days, unless it will take extra time to address the concerns of the reviewers, in which case we would appreciate an expected resubmission date by email to plosgenetics@plos.org.

We are sorry that we cannot be more positive about your manuscript at this stage. Please do not hesitate to contact us if you have any concerns or questions.

Yours sincerely,

Jesse R Lasky

Guest Editor

PLOS Genetics

Kelly Dyer

Section Editor

PLOS Genetics

Dear Dr. Kelly,

I have now received three reviews of your manuscript. Overall, the reviewers appreciated the study for the depth and scope of quantitative genetics findings on gene expression. The study is well designed and the dataset and analyses are rich.

However, perhaps reflecting the complexity of the topic, there are many points of clarification that the reviewers emphasized. Next, as one reviewer noted, the connection between your findings and underlying evolutionary processes is particularly interesting. There was some discussion of this in the manuscript, but the reviewer and I felt it could be sharpened and strengthened. Additionally, reviewers noted that the provided code appeared to be missing important analyses.

I look forward to receiving a revised version of the manuscript.

Sincerely,

Jesse Lasky

Reviewer's Responses to Questions

**Comments to the Authors:**

Reviewer #1: This manuscript investigates genetic variation in gene expression in a population of Mimulus guttatus. It uses a multi-parental mapping population to avoid some of the pitfalls of GWAS-based approaches and more accurately estimate properties of gene expression variation. The manuscript shows that almost every gene has a cis-eQTL, many with multiple alleles segregating in the population. While there were detectable trans-eQTLs for many genes, they still did not explain most of the trans-expression variation, and they had stronger dominance and were at lower allele frequencies than cis-eQTL. In addition, the manuscript demonstrates how trans-eQTLs can contribute to coexpression and shows that mapping expression PCs can detect eQTLs.

This manuscript’s results fill in large gaps in our basic understanding of the genetics of gene expression. There are lots of hypotheses about how cis- and trans- eQTLs might differ from each other in terms of dominance, selection, diversity, and effect size but, until now, we haven’t had very much solid data on this from plant species. If anything, I think the manuscript undersells how important these findings will be for future work in the evolution of gene expression. The methods and approaches will also be very useful for future studies. In general, the manuscript is well-written and clear, although I have a few comments about places where additional explanation or writing might increase clarity, as well as some other suggestions to help the reader. I do not have any major concerns about the research itself.

Comments (in order of where they relate to the manuscript).

It would be helpful to define ‘cis’ regulation when it is first mentioned on line 64.

Fig 1C – I was a bit thrown off by the scale of pairwise relatedness being from 0-2 instead of 0-1. Could you briefly explain the measure of relatedness in the legend instead of just in the methods?

Line 151 – it wasn’t clear to me what “the cis genotype” means here.

Line 185 – Could you add in either the number of genes with a cis-eQTL in the combined analysis or the percent of genes with cis eQTLs in the cross-specific analysis to make it easier to compare the two values?

Thinking of those who might skim this paper so they can cite the result “X% if genes have a cis-eQTL”, could you be clearer about how this percent is out of the genes that were investigated, not all genes. It could be helpful to add in that this means that at least Y% of all genes have a cis-eQTL where Y is the number of genes with a mapped cis-eQTL divided by the total number of genes.

Fig 2 – can you explain the differences in the colors of the dots?

I couldn’t figure the distance cut-off for whether a QTL was cis or trans. Can you include this info more prominently?

Line 221 – could you replace “is usually” with a value for how often the allele that increases expression across genotypes has higher allele-specific expression in heterozygotes?

The analysis described on 232-236 took me a while to figure out – would it be possible to add a figure to explain how the dominance of a cis-eQTL affects allele-specific expression? One possibility would be plotting d on the x axis and the residual of the regression of the alt-allele frequency on a on the y axis? (this is not a requirement, and could possibly make things more confusing).

Line 288 – could you remind the readers of the average cis contribution to Vg here?

Figure 4B be simplified by reporting the frequency of the most common allele, so for example columns [1,3,6] and [2,2,6] would be collapsed.

The reasoning of the paragraph starting on line 303 was a bit hard to follow. First, can you define H? Second, I think a bit more explanation about the comparison between the cross-specific and combined analysis would be helpful.

On line 369, it would be useful to have a bit more evidence that the pcQTLs overlap transeQTL hotspots, maybe in a supplemental figure or even reporting a value of how many pcQTLs are within 2 centimorgans of a hotspot or something.

Line 393 – the results about the relationship between sequence diversity at or around a gene and high Vg(cis) reminded me a bit of the results of Kremling et al. 2018 (https://doi.org/10.1038/nature25966), so it could be interesting to cite this and related papers here.

Could you explain more about why different procedures were used for estimating read counts in the cross-specific and combined analyses? (line 654ish)

In the methods, could you add what programs or packages were used to fit the various models? I was a bit confused about why detailed code was given for the false discovery rate analysis (starting line 735) but not for other aspects of the statistical analyses or the simulations in Appendix 1.

In the first set of null simulations, how was environmental variation in expression generated? (line 777)

Reviewer #2: The study by Veltsos and Kelly assembles a truly stunning dataset comprising 10 F2 populations of the model ecological genetic species Mimulus gutatus, with 10 genetically diverse individuals collected from a single natural population each crossed to a common maternal line. This type of multi-parent recombinant population is perfectly suited for addressing classic quantitative genetic parameters such as the genetic variance and covariance, dominance, and genotype by environment interaction effects on traits. In the current study, the authors utilize these F2 individuals to estimate transcript abundances using RNASequencing – eQTL-- providing an unprecedented view into the genetic architecture of gene expression in plants. This work builds on, and substantially extends, a recent paper by this same group which used an association mapping approach to study the genetic architecture of gene expression using a broader sample of natural genotypes. The two studies strongly complement each other, and the authors do an excellent job of addressing similarities and differences in their results.

On the whole, I find this study remarkable and groundbreaking. It should serve as a template and benchmark for future eQTL studies, and deepens our understanding of the quantitative genetics of complex traits. The methods are well-executed, applying modern tools from quantitative genetics, genomics, and transcriptomics to classic questions in the evolution of complex traits.

Below are several suggestions to improve the manuscript. Most deal with presentation and interpretation.

Somewhat major points

The first paragraph (at Line 52) has too many un-expanded questions which lack context. I think this paragraph could be better utilized to make the case – in prose – that A. gene expression is a quantitative trait. B. there is a vast body of literature and theory on the evolution of quantitative traits. C. Many key processes in evolution can be studied by a modern, genome-enabled, perspective of A & B. In my view, this paper will be greatly appreciated by the quantitative genetics community but may be missed by the gene regulation and population genomics communities. Offering a gentle introduction to these latter communities would greatly increase the impact of this study.

Line 64 – need a reference for cis-regulatory siting of expression variance. The following line “The priority of the cis-regulatory region…” also needs a reference, and a more general introduction to this important earlier observation would help the general reader.

Line 138-139. Trying to get my head around this point. Wouldn’t limited recombination, which limits mapping resolution, also make identifying the location of cis-eQTL harder? We know that the marker is linked to the causal variant, but with high LD it could be pretty far away even if technically in cis (linked)?

Line 157 – Support for the claim that the “crossing design produced the high variance in relatedness necessary for accurate estimation”? I don’t doubt this to be true, but this is written as a strong claim and I think some metric is warranted.

Line 201 – similarly to preceding comment “The point estimates are remarkably consistent;” are they consistent in some way that can be expressed with a metric?

Line 215 – I’d love to see a distribution of how “local” local eQTLs are. Like how far, on average, is a marker from the transcript whose variance it explains?

The “Concluding remarks” include real gems of interpretation that I will return to frequently in the coming weeks. But some of this is written too colloquially. The first sentence, second clause, is missing words. And “etc.” at line 510 is too casual. Even for PLoS.

I may have missed it, but is a link given for the scripts used in the bioinformatic and quantitative genetic analyses?

Minor points

Line 52 – first use of “RNASeq” should probably spell this out. Better yet, I’d lead with “Gene expression is a quantitative trait.” No need for the qualification.

Line 72 – “cis-regulatory molecules” What the heck is that? Do you mean transcription factors? Say so.

Line 566 – how were tissues disrupted? With a bead beater or similar?

Reviewer #3: In this study, Veltsos and Kelly address the dynamics of eQTLs and their contribution to genetic variation in Mimulus guttatus via a multiparent mapping population. Ultimately, this work highlights the quantitative landscape of gene expression in leaves of a natural population of Mimulus. The authors showed that in M. guttatus a majority of the genetic variation in gene expression (two-thirds) arises from allelic diversity amongst cis eQTLs for the Iron Mountain population. On the other hand, trans eQTLs show stronger signals of dominance compared to cis eQTLs and are correlated with multiple expressed genes, most likely functioning as network regulators. The authors have generated an impressively large and novel transcriptomic dataset that will be valuable to the community. We do, however, have some comments:

1) Given the strong cis-eQTL effects on gene expression it might be interesting to see if allelic series lead to multimodal expression patterns (see for example Hsieh et al, Genome Biology, 2007: Mixture modeling of transcript abundance classes in natural populations.

2) The figure legends could be greatly enhanced with increased explanations for each panel. Currently, the figures and legends alone are not sufficient to interpret all of the results without digging into the methods and results. For example, Figure 3D, what does d/a represent?

3) The authors could consider to present read counts on the y axis of several figure panels as relative read counts, as the number of genes expressed varies widely between species.

4) The study compares their findings of high allelic diversity of cis eQTLs to Liu et al., 2019. This is a relevant study that compares the contribution of cis/trans eQTLs for several species and we appreciate the broad scope. However, this comparison contains only studies on animals and yeast. If cis/trans comparisons have been made in plants, these would be beneficial for the reader.

For example, work from Nathaniel Street and colleagues on selective constraint on eQTL in poplar might be relevant (Mähler et al, PLoS Genetics, 2017: Gene co-expression network connectivity is an important determinant of selective constraint) as might eQTL studies on RIL populations in Arabidopsis by Dan Kliebenstein, Maarten Koornneef, and Thomas Juenger and their colleagues (e.g., https://doi.org/10.1073/pnas.0610429104, https://doi.org/10.1105/tpc.113.115352 and https://doi.org/10.1534/genetics.106.064972).

5) Given this manuscript’s focus on G matrix evolution there appear to be some relevant references missing, for example some of the work by Mark Blows and colleagues on G matrix evolution in Drosophila serrata (see for example Blows et al, American Naturalist, 2015: The phenome-wide distribution of genetic variance).

6) The order in which figure panels are introduced in the text is not always congruent with the actual order of figure panels.

7) In addition, it appears that the supplementary tables are not listed in numerical order according to appearance in the manuscript. For example, the first supplementary table listed is #8.

8) Formatting inconsistency: Most of the text uses the format cis/trans eQTLs. However, several times in the manuscript the format cis/trans-eQTL is used (lines 69, 690, 692, 806).

9) Lines 185 and 186: how do the combined analysis and cross-specific analysis compare to one another in terms of the numbers of cis-eQTLs identified?

10) Lines 207 and 208: VE appears high. Were any steps taken to minimize or control for environmental variation? For example, to control for circadian rhythm effects, how much time was taken for sampling within a cohort of 500 plants? Was each cohort sampled at the same or a different time of day? And were families present across multiple cohorts or kept within single cohorts? The latter might influence the combined versus the cross-specific analyses?

11) Lines 272 and 273: how do the combined analysis and cross-specific analysis compare to one another in terms of the numbers of trans-eQTLs identified?

12) Lines 292 and 293: I would say that the distribution for trans-eQTL is quite even, but skewed toward [4,6] for cis-eQTL?

13) Line 512: “at the Iron Mountain location” – this is one instance where I think that the paper could provide a little more introduction to the study system. Many potential readers will not know what “Iron Mountain” refers to?

14) Typo: Line 41: “While associations studies routinely characterize genetic…”

15) Typo: Line 60: “…genetic and environmental covariances to co-expression…”

16) Typo: Line 114, 556: denovo, requires a space, de novo, as used elsewhere in the text.

17) Typo: Line 161-162: “…can have loci identical by descent through both parents.”

18) Typo: Line 163: “…formation that will makes siblings more/less…”

19) Typo: Line 180: “We chose Vg[a] to estimate the…” (missing a space)

20) Typo: Line 200: “…for cis-regulatory variation is much stronger…”

21) Typo: Line 267: “Cis eQTl…”

22) Typo: Line 272: “cis eQTLs, trans eQTLs are abundant…”

23) Typo: Line 282: “ …98% of the trans eQTLs/affected gene pairs were..” (extra space)

24) Typo: Line 292: “…latter exhibit a more even distribution…”

25) Typo: Line 404: “…intermediate frequencies relative to the overall genomic…”

26) Typo: Line 414: “Multi-parental mapping populations in both plants…”

27) Typo: Line 426: “…many trans-SNPs where the minor allele…”

28) Typo: Line 466: “…own transcription by binding to their own promoter region.”

29) Typo: Line 501: “…for the co-expression patterns that determine…”

30) Type: Line 529: “From one generation to the next,…” (extra space after period)

**Have all data underlying the figures and results presented in the manuscript been provided?**

Reviewer #1: Yes

Reviewer #2: Yes

Reviewer #3: Yes

PLOS authors have the option to publish the peer review history of their article (what does this mean?). If published, this will include your full peer review and any attached files.

Reviewer #1: No

Reviewer #2: No

Reviewer #3: No

---

## [Decision Letter · Decision Letter 1]

23 Mar 2024

Dear Dr Kelly,

We are pleased to inform you that your manuscript entitled "The quantitative genetics of gene expression in Mimulus guttatus" has been editorially accepted for publication in PLOS Genetics. Congratulations!

Yours sincerely,

Jesse R Lasky

Guest Editor

PLOS Genetics

Kelly Dyer

Section Editor

PLOS Genetics

Comments from the reviewers (if applicable):

The reviewers and I were all happy to see the revised version of this manuscript. It is an exciting contribution and our pleasure to accept the manuscript.

Sincerely,

Jesse Lasky

Reviewer's Responses to Questions

**Comments to the Authors:**

Reviewer #1: I appreciate the authors’ attention and responses to all of the comments. They have made a number of changes that make the manuscript easier to read and understand. I have no further comments, but want to emphasize that I'm really enthusiastic about this study and think it will be a great contribution to the literature on gene expression variation!!!

Reviewer #2: The authors have done a nice job addressing my earlier comments. I have nothing to add for the current draft

Reviewer #3: In my view, the authors have responded well to feedback from the editor and all three reviewers on the initial version of the manuscript. I think the paper is impressive and I do not have any further qualms about it.

**Have all data underlying the figures and results presented in the manuscript been provided?**

Reviewer #1: None

Reviewer #2: Yes

Reviewer #3: Yes

PLOS authors have the option to publish the peer review history of their article (what does this mean?). If published, this will include your full peer review and any attached files.

Reviewer #1: No

Reviewer #2: **Yes: **David L Des Marais

Reviewer #3: **Yes: **Simon C. Groen

**Data Deposition**

http://datadryad.org/submit?journalID=pgenetics&manu=PGENETICS-D-23-01298R1

**Press Queries**

---

## [Editor Report · Acceptance letter]

3 Apr 2024

PGENETICS-D-23-01298R1 

The quantitative genetics of gene expression in Mimulus guttatus 

Dear Dr Kelly, 

We are pleased to inform you that your manuscript entitled "The quantitative genetics of gene expression in Mimulus guttatus" has been formally accepted for publication in PLOS Genetics! Your manuscript is now with our production department and you will be notified of the publication date in due course.

With kind regards,

Zsofia Freund

PLOS Genetics

On behalf of:
